



# Light absorption properties of aerosols over Southern West Africa

Cyrielle Denjean[1], Thierry Bourrianne[1], Frederic Burnet[1], Marc Mallet[1], Nicolas Maury[1], Aurélie Colomb[2], Pamela Dominutti[2,*], Joel Brito[2,**], Régis Dupuy[2], Karine Sellegri[2], Alfons Schwarzenboeck[2], Cyrille Flamant[3], Peter Knippertz[4]

[1]CNRM, Université de Toulouse, Météo-France, CNRS, Toulouse, France
[2]LaMP, Université de Clermont Auvergne, Clermont-Ferrand, France
[3]LATMOS/IPSL, Sorbonne Université, UVSQ, CNRS, Paris, France
[4]Institute of Meteorology and Climate Research, Karlsruhe Institute of Technology, Karlsruhe, Germany
* Now at Wolfson Atmospheric Chemistry Laboratories, Department of Chemistry, University of York, YO10 5DD- York, UK
** Now at: IMT Lille Douai, Université de Lille, SAGE, Lille, France

*Correspondence to : Cyrielle Denjean (cyrielle.denjean@meteo.fr)*

**Abstract.** Southern West Africa (SWA) is an African pollution hotspot but a relatively poorly sampled region of the world. We present an overview of *in-situ* aerosol optical measurements collected over SWA in June and July 2016 as part as the DACCIWA (Dynamics-Aerosol-Chemistry-Clouds Interactions in West Africa) airborne campaign. The aircraft sampled a wide range of air masses, including anthropogenic pollution plumes emitted from the coastal cities, long-range transported biomass burning plumes from Central and Southern Africa and dust plumes from the Sahara and Sahel region, as well as mixtures of these plumes. The specific objective of this work is to characterize the regional variability of the vertical distribution of aerosol particles and their spectral optical properties (single scattering albedo: *SSA*, asymmetry parameter, extinction mass efficiency, scattering Ångström exponent and absorption Ångström exponent: *AAE)*. First findings indicate that aerosol optical properties in the planetary boundary layer were dominated by a widespread and persistent biomass burning loading from the Southern Hemisphere. Despite a strong increase of aerosol number concentration in air masses downwind of urban conglomerations, spectral *SSA* were comparable to the background and showed signatures of the absorption characteristics of biomass burning aerosols. In the free troposphere, moderately to strongly absorbing aerosol layers, dominated by either dust or biomass burning particles, occurred



occasionally. In aerosol layers dominated by mineral dust particles, *SSA* varied from 0.81 to 0.92 at
550 nm depending on the variable proportion of anthropogenic pollution particles externally mixed
with the dust. Biomass burning aerosol particles were significantly more light absorbing than those
previously measured in other areas (e.g. Amazonia, North America) with *SSA* ranging from 0.71 to
0.77 at 550 nm. The variability of *SSA* was mainly controlled by variations in aerosol composition
rather than in aerosol size distribution. Correspondingly, values of *AAE* ranged from 0.9 to 1.1,
suggesting that lens-coated black carbon particles were the dominant absorber in the visible range
for these biomass burning aerosols. Comparison with literature shows a consistent picture of
increasing absorption enhancement of biomass burning aerosol from emission to remote location
and underscores that the evolution of *SSA* occurred a long time after emission.
The results presented here build a fundamental basis of knowledge about the aerosol optical
properties observed over SWA during the monsoon season and can be used in climate modelling
studies and satellite retrievals. In particular and regarding the very high absorbing properties of
biomass burning aerosols over SWA, our findings suggest that considering the effect of internal
mixing on absorption properties of black carbon particles in climate models should help better
assessing the direct and semi-direct radiative effects of biomass burning particles.

**1. Introduction**
Atmospheric aerosols play a crucial role in the climate system by altering the radiation budget
through scattering and absorption of solar radiation and by modifying cloud properties and
lifetime. Yet considerable uncertainties remain about the contribution of both natural and
anthropogenic aerosol to the overall radiative effect *(Boucher et al., 2013)*. Large uncertainties are
related to the complex and variable properties of aerosol particles that depend on the aerosol
source and nature as well as on spatial and temporal variations. During transport in the
atmosphere, aerosol particles may undergo physical and chemical aging processes altering the
composition and size distribution and henceforth the optical properties and radiative effects. The
capability of reproducing this variability in climate models represents a real challenge *(Myhre et*
*al., 2013; Stier et al., 2013; Mann et al., 2014)*. Therefore, intensive experimental observations in
both aerosol source and remote areas are of paramount importance for constraining and evaluating
climate models.

Key parameters from a climate perspective are the aerosol vertical distribution and respective
spectral optical properties. Radiative transfer codes commonly incorporated in climate models and
in satellite data retrieval algorithms use single scattering albedo (*SSA*), extinction mass coefficient



(*MEC)* and asymmetry factor (*g*) as input parameters. These parameters depend on the aerosol size
distribution, the real and imaginary parts of the refractive index (*m-ik)*, and the wavelength of
incident light, $\lambda$. The knowledge of the vertical distribution of these fundamental parameters is
crucial to accurately estimate the direct and semi-direct radiative effects of aerosols as well as the
vertical structure of atmospheric heating rates resulting from absorption by particles. Above
information is also required to retrieve aerosol properties (aerosol optical depth, size distribution)
from remote sensing data.

Southern West Africa (SWA) is one of the most climate-vulnerable region in the world , where the
surface temperature is expected to increase by ~3°K at the end of the century (2071-2100) in the
Coupled Model Intercomparison Project Phase 5 (CMIP5) *(Roehrig et al., 2013)*. It is
characterized by a fast-growing population, industrialization and urbanization *(Liousse et al.,*
*2014)*. This is particularly the case along the Guinea Coast where several already large cities are
experiencing rapid growth *(Knippertz et al., 2015a)*. Despite these dramatic changes, poor
regulation strategies of traffic, industrial and domestic emissions lead to a marked increase of
anthropogenic aerosol loading from multiple sources including road traffic, industrial activities,
waste burning, ship plumes, domestic fires, power plants, etc. Tangible evidence for regional
transport of anthropogenic pollutants associated with urban emissions has altered air pollution
from a local issue to a regional issue and beyond *(Deetz et al., 2018; Deroubaix et al. 2019)*. This
is particularly the case during summer when land-sea breeze systems can develop and promote the
transport of pollutants inland, away from the urbanized coastal strip of SWA *(Flamant et al.,*
*2018a)*. In addition to this anthropogenic regional pollution, SWA is impacted by a significant
import of aerosols from remote sources. Biomass burning mainly from vegetation fires in Central
Africa are advected to SWA in the marine boundary layer and aloft *(Mari et al., 2008; Menut et al.*
*2018; Haslett et al., 2019)*. The nearby Sahara desert and the Sahel are large sources of natural
wind-blown mineral dust aerosol throughout the year with a peak in springtime *(Marticorena and*
*Bergametti, 1996)*. Biomass burning, dust and anthropogenic pollution aerosols can be mixed
along their transport pathways *(Flamant et al., 2018a; Deroubaix et al. 2019)*, resulting in
complex interactions between physical and chemical processes and even meteorological
feedbacks.

In West Africa, most of the aerosol–radiation interaction studies focused on optical properties of
dust and biomass burning aerosols in remote regions far from major sources of anthropogenic
pollution aerosol. They include ground-based and airborne field campaigns such as DABEX (Dust



and Biomass Experiment, *Haywood et al., 2008)*, AMMA (Analysis Multidisciplinary of African
Monsoon, *Lebel et al., 2010)*, DODO (Dust Outflow and Deposition to the Ocean, *McConnell et
al., 2008)*, SAMUM-1 and SAMUM-2 (Saharan Mineral Dust Experiment, *Heintzenberg, 2009;
Ansmann et al., 2011)* and AER-D (AERosol Properties – Dust, *Ryder al. 2018)*. These projects
concluded that the influence of both mineral dust and biomass burning aerosols on the radiation
budget is significant over West Africa, implying that meteorological forecast and regional/global
climate models should include their different radiative effects for accurate forecasts and climate
simulations. Over the Sahel region, *Solmon et al. (2008)* have highlighted the high sensitivity of
mineral dust optical properties to precipitation changes at a climatic scale. However, the optical
properties of aerosols particles in the complex chemical environment of SWA are barely studied.
This is partly due to the historically low level of industrial developments of the region. Motivated
by the quickly growing cities along the Guinea Coast, the study of transport, mixing, and feedback
processes of aerosol particles is therefore very important for better quantification of aerosol
radiative impact at the regional scale and improvement of climate and numerical weather
prediction models.

In this context, the DACCIWA (Dynamics-Aerosol-Chemistry-Clouds Interactions in West Africa,
*Knippertz et al., 2015b*) campaign, designed to characterize both natural and anthropogenic
emissions over SWA, provides important and unique observations of aerosols in a region much
more affected by anthropogenic emissions than previously thought. A comprehensive field
campaign took place in June–July 2016 including extensive ground-based *(Kalthoff et al., 2018)*
and airborne measurements *(Flamant et al., 2018b)*. In this study, we present an overview of *in-
situ* airborne measurements of the vertical distribution of aerosol particles and their spectral optical
properties acquired with the ATR-42 French research aircraft over the Guinea Coast.

Section 2 presents the flight patterns, instrumentation and data analysis. Section 3 provides an
overview of the aerosol microphysical and optical properties. The impact of aging and mixing
processes on aerosol optical properties is discussed in section 4 before conclusions are presented in
section 5.



## 2. Methodology

### 2.1. ATR-42 measurements overview

This analysis focuses on flight missions conducted by the ATR-42 aircraft of SAFIRE (Service des Avions Français Instrumentés pour la Recherche en Environnement - the French aircraft service for environmental research) over the Gulf of Guinea and inland. A full description of flight patterns during DACCIWA is given in *Flamant et al. (2018b)*. Here we present results from 15 flights focused on the characterization of anthropogenic pollution, dust and biomass burning plumes. The flight tracks are shown in Figure 1 and a summary of flight information is provided in Appendix 1. The sampling strategy generally consisted of two parts: first, vertical soundings were performed from 60 m up to 8 km above mean sea level (amsl) to observe and identify interesting aerosol layers. Subsequently, the identified aerosol layers were probed with the *in-situ* instruments by straight levelled runs (SLR) at fixed flight altitudes.

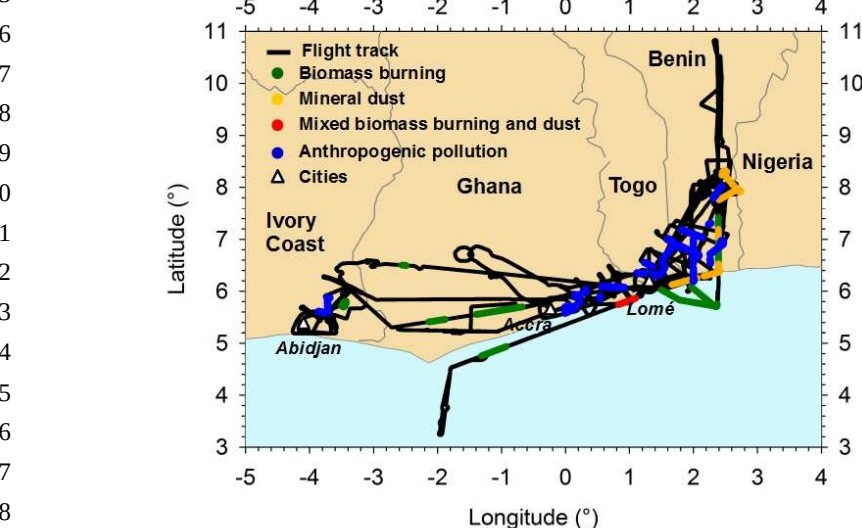

**Figure 1. Tracks of the 15 flights analyzed in this study. The colors indicate aircraft flight levels sampling layers dominated by biomass burning (green), mineral dust (orange), mixed dust-biomass burning (red) and anthropogenic pollution particles (blue).**

The ATR-42 aircraft was equipped with a wide variety of instrumentation performing gas and aerosol measurements. The measured meteorological parameters include temperature, dew point temperature, pressure, turbulence, relative humidity, as well as wind speed and direction. Gas phase species were sampled through a rear facing ¼ inch Teflon tube. Carbon monoxide (CO) was





measured using ultra-violet and infrared analysers (PICARRO). The nitric oxide (NO) and
nitrogen dioxide ($NO_2$) measurements were performed using an ozone chemiluminescence
instrument (Thermo Environmental Instrument TEi42C with a Blue Light Converter for the NO2
conversion). On-board aerosol instruments sampled ambient air via stainless steel tubing through
the Community Aerosol Inlet (CAI). This is an isokinetic and isoaxial inlet with a 50 % sampling
efficiency for particles with a diameter of 5 μm *(Denjean et al., 2016)*.

The total number concentration of particles larger than 5 nm (*Ntot*) was measured by a
condensation particle counter (CPC model MARIE built by University of Mainz). The aerosol size
distribution was measured using an ultra-high sensitivity aerosol spectrometer (UHSAS, DMT), a
custom-built scanning mobility sizer spectrometer (SMPS) and an optical particle counter (OPC,
GRIMM model 1.109). Instrument calibration was performed with PSL nanospheres and oil
particles size-selected by a differential mobility analyser (DMA) for diameters from 90 nm to 20
μm. The SMPS data acquisition system failed after two-third of the campaign and could not be
repaired. We found the UHSAS to show false counts in the diameters below 100 nm. Therefore,
these channels were disregarded in the data analysis.

The particle extinction coefficient ($\sigma_{ext}$) at the wavelength of 530 nm was measured with a cavity
attenuated phase shift particle light extinction monitor (CAPS-PMex, Aerodyne Research). The
particle scattering coefficients ($\sigma_{scat}$) at 450, 525 and 700 nm were measured using a TSI 3-
wavelength nephelometer. The absorption coefficients ($\sigma_{abs}$) at 467, 520 and 660 nm were
measured by a Radiance Research Particle Soot Absorption Photometer (PSAP). The PSAP
measures changes of filter attenuation due to the collection of aerosol deposited on the filter, which
were corrected for the scattering artifacts according to the *Virkkula (2010)* method. Prior to the
campaign, the CAPS was evaluated against the combination of the nephelometer and the PSAP. An
instrument intercomparison was performed with purely scattering ammonium sulfate particles and
with strongly absorbing black carbon particles (BC). Both types of aerosol were generated by
nebulizing a solution of the respective substances and size-selected using a DMA. For instrument
intercomparison purposes, $\sigma_{ext}$ from the combination of nephelometer and PSAP was adjusted to
that for 530 nm by using the scattering and absorption Ångstrom exponent (*SAE* and *AAE*,
respectively). The instrument evaluation showed an excellent accuracy of the CAPS measurements
by comparison to the nephelometer and PSAP combination.


## 2.2. Ancillary products

In order to determine the history of air masses prior aircraft sampling, backward trajectories and satellite images were used. The trajectories were computed using the Hybrid Single Particle Lagrangian Integrated Trajectory Model (HYSPLIT) and the National Centers for Environmental Prediction (NCEP) Global Data Assimilation System (GDAS) data with 0.5° horizontal resolution for sequences and times of interest. We compared the backward trajectory heights with information of fire burning times (e.g. MODIS Burnt Area Product) and dust release periods (e.g. Meteosat Second Generation (MSG) dust RGB composite images) to assess the aerosol source regions of the investigated air masses. The air masses represented by the trajectory are assumed to obtain their aerosol loading from source regions when the trajectory passes over regions with significant dust activation and/or fire activity at an altitude close to the surface. Trajectory calculations with slightly modified initial conditions with respect to the arrival time, location and altitude were performed to check the reliability of the location of source regions. Uncertainties in this approach, caused by unresolved vertical mixing processes, and by general uncertainties of the trajectory calculations are estimated to be in the range of 15–20 % of the trajectory distance *(Stohl et al., 2002)*.

## 2.3. Data analysis

In the following, extensive aerosol parameters (concentrations, scattering, absorption and extinction coefficients) are converted to standard temperature and pressure (STP) using T = 273 K and P = 1013.25 hPa. The STP concentration data correspond to mixing ratios, which are independent of ambient pressure and temperature during the measurement. In the analysis, the data were averaged over sections of SLR with homogeneous aerosol conditions outside of clouds.

### 2.3.1. Derivation of aerosol microphysical and optical properties

Appendix 2 and Table 1 show the iterative procedure and the equations used to calculate the aerosol microphysical and optical parameters as briefly explained below.

The particle number concentration in the coarse mode ($N_{coarse}$) was calculated by integrating the OPC size distributions over the range 1 to 5 µm. The number concentration of particles in the fine mode ($N_{fine}$) was obtained as the difference between total number concentration ($N_{tot}$ particle diameter range above 5 nm) measured by the CPC and $N_{coarse}$.



For optical calculations, the 3λ-$\sigma_{abs}$ from the PSAP were adjusted at the 3 wavelengths measured
by the nephelometer using the *AAE* calculated from the 3λ measured $\sigma_{abs}$. Once $\sigma_{scat}$ and $\sigma_{abs}$
obtained at the same wavelength, an optical closure study estimated the complex refractive index
based on optical and size data. Optical calculations were performed using Mie theory for spherical
particles by varying stepwise the real part of the complex refractive index (*m*) from 1.33 to 1.60
and the imaginary part of the complex refractive index (*k*) from 0.000 to 0.080. Given that the size
distribution measured by the UHSAS and the OPC depends on *n*, the optical-to-geometrical
diameter conversion was recalculated at each iteration based on the assumed *n*. The resulting
number size distributions from SMPS, UHSAS and OPC were parameterized by fitting four log-
normal distributions and used as input values in the optical calculations. *m* and *k* were fixed when
the best agreement between calculated values of $\sigma_{scat}$ and $\sigma_{abs}$ and measurements was found. Once
*n* and *k* were obtained at 3λ, we estimated the following optical parameters:
- *SAE* depends on the size of the particles. Generally, it is lower than 0 for aerosols dominated by
coarse particles, such as dust aerosols, but it is higher than 0 for fine particles, such as
anthropogenic pollution or biomass burning aerosol *(Seinfeld and Pandis, 2006; Schuster et al.,*
*2006).*
- *AAE* provides information about the chemical composition of atmospheric aerosols. BC absorbs
radiation across the whole solar spectrum with the same efficiency, thus it is characterized by *AAE*
values around 1. Conversely, mineral dust particles show strong light absorption in the blue to
ultraviolet spectrum leading to *AAE* values up to 3 *(Kirchstetter et al., 2004; Petzold et al., 2009).*
- *SSA* describes the relative importance of scattering and absorption for radiation. Thus, it indicates
the potential of aerosols for cooling or warming the lower troposphere.
- *g* describes the probability of radiation to be scattered in a given direction. Values of *g* can range
from −1 for entirely backscattered light to +1 for complete forward scattering light.
- *MEC* represents the total light extinction per unit mass concentration of aerosol. The estimates of
*MEC* assume mass densities of 2.65 g cm$^3$ for dust aerosol, 1.35 g cm$^3$ for biomass burning
aerosol, 1.7 g cm$^3$ for anthropogenic aerosol and 1.49 g cm$^3$ for background aerosol *(Hess et al.,*
*1998; Haywood et al., 2003a).*




| Aerosol parameters | Symbol | $\lambda$ (nm) | Method |
|---|---|---|---|
| **Aerosol microphysical properties** | | | |
| Total number concentration | $N_{tot}$ | - | Measured by a CPC in the particle diameter range above 5 nm |
| Number concentration in the coarse mode | $N_{coarse}$ | - | GRIMM size distributions integrated on the range 1 to 5 µm. |
| Number concentration in the fine mode | $N_{fine}$ | - | Difference $N_{tot}$ and $N_{coarse}$. |
| Number size distribution | dN/dlogDp | - | $dN/dlogDp = \Sigma_{i=1}^{4} (N_{tot,i}\ exp(-(logD_p - logD_{p,g,i})^2/(2\ log\ \sigma_i))/(\sqrt{2}\ log\ \sigma_i))$ with $N_{tot,i}$ the integrated number concentration, $D_{p,g,i}$ the geometric median diameter and $\sigma_i$ geometric standard deviation for each mode i |
| Volume size distribution | dV/dlogDp | - | $dV/dlogDp = \Sigma_{i=1}^{4} (N_{tot,i}\ D_p^3\ \pi/6\ exp(-(logD_p - logD_{p,g,i})^2/(2\ log\ \sigma_i))/(\sqrt{2}\ log\ \sigma_i))$ |
| **Aerosol optical properties** | | | |
| Scattering coefficient | $\sigma_{scat}$ | 450, 550, 700 | Measured by the nephelometer and corrected for truncator error |
| Absorption coefficient | $\sigma_{abs}$ | 467, 520, 660 | Measured by the PSAP and corrected for filter based artefacts |
| Extinction coefficient | $\sigma_{ext}$ | 530 | Measured by the CAPS |
| Scattering Ångström exponent | SAE | 450 to 700 | Calculated from the nephelometer measurements : $SAE = -ln\ (\sigma_{scat}(450)/\sigma_{scat}(700))\ /\ ln\ (450/700)$ |
| Absorption Ångström exponent | AAE | 440 to 660 | Calculated from the PSAP measurements : $AAE = -ln(\sigma_{abs}(467)/\sigma_{abs}(660))\ /\ ln\ (467/660)$ |
| Complex refractive index | n | 450, 550, 660 | Inversion closure study using Mie theory (Fig. A2) $n(\lambda) = m(\lambda)-ik(\lambda)$ |
| Single scattering albedo | SSA | 450, 550, 660 | Inversion closure study using Mie theory (Fig. A2) $SSA(\lambda) = \sigma_{scat}(\lambda)/\sigma_{ext}(\lambda)$ |
| Mass extinction efficiency | MEC | 450, 550, 660 | Inversion closure study using Mie theory (Fig. A2) $MEC(\lambda) = \sigma_{ext}(\lambda)/C_m$ with $C_m$ the aerosol mass concentration |
| Asymmetry parameter | g | 450, 550, 660 | Inversion closure study using Mie theory (Fig. A2) $g(\lambda) = 1/2\ \int_0^\pi cos(\Theta)sin(\Theta)P(\Theta,\lambda)d(\Theta)$ with $P(\Theta,\lambda)$ the scattering phase function and $\Theta$ the scattering angle. |

**Table 1. Aerosol microphysical and optical properties derived in this work**



### 2.3.2. Classification of aerosols plumes

Data were screened in order to isolate plumes dominated by anthropogenic pollution from urban emissions, biomass burning and mineral dust particles, resulting in a total number of 19, 12 and 8 genuine plume interceptions, respectively, across the 15 flights. Identification of the plumes was based on a combination of CO and NOx (sum of NO and $NO_2$) concentrations, as well as *AAE* and *SAE* that have been shown to be good parameters for classifying aerosol types *(Kirchstetter et al., 2004; Petzold et al., 2009)*. The classification was then compared with results from the back trajectory analysis (Figure 2) and satellite images described in section 2.2. The guidelines for classification are as follows:

- *Anthropogenic pollution*: *SAE* was beyond threshold 0, indicating a large number fraction of small particles in urban plumes, and CO and NOx concentrations 2 times higher than the background concentrations. During the DACCIWA campaign background CO and NOx values were around 180 ppb and 0.28 ppb, respectively. The trajectories show large differences in the flow patterns and source regions with urban plumes originating from the polluted cities of Lomé, Accra and Abidjan. The aircraft sampling over land mostly followed the north-eastward direction (Figure 2d).

- *Biomass burning*: The criteria are the same as for urban pollution plumes except that trajectories track theses plumes back to active fire hotspots as observed by MODIS and the ratio NOx to CO was set below 1. CO and NOx are byproducts of combustion sources but CO is preserved longer along the plume when compared with NOx, which makes the ratio NOx to CO a good indicator for distinguishing fresh anthropogenic pollution plumes from biomass burning plumes transported over long distances *(Wang et al., 2002; Silva et al., 2017)*. During this time of the year, most of the forest and grassland fires were located in Central and Southern Africa (Figure 2a).

- *Mineral dust*: *AAE* higher than 1 indicates a large mass fraction of mineral dust and a *SAE* below 0 indicate a high effective particle diameter. The source region of the dust loaded air masses was located in the Saharan desert and in the Sahel (Figure 2b).

- *Dust and biomass burning mixing:* Combining remote sensing observations and model simulations, *Flamant et al. (2018a)* identified a biomass burning plume mixed with mineral dust. This agrees well with the measured *AAE* of 1.2 and *SAE* of 0.3 observed in this layer. *Menut et al. (2018)* have shown that one of the transport pathways of biomass burning aerosols from Central Africa was associated with northward advection towards Chad and then westward displacement linked to the African Easterly Jet. The plume originated from a broad active biomass burning area including Gabon, the Republic of Congo and the Democratic Republic of Congo and passed over





areas with strong dust emissions further north within 1–3 days before being sampled by the aircraft
(Figure 2c).
- *Background:* We refer to background conditions as an atmospheric state in the boundary layer
without the detectable influence of local anthropogenic pollution sources. Most back trajectories
originated from the marine atmosphere and coastal areas south of the sampling area.

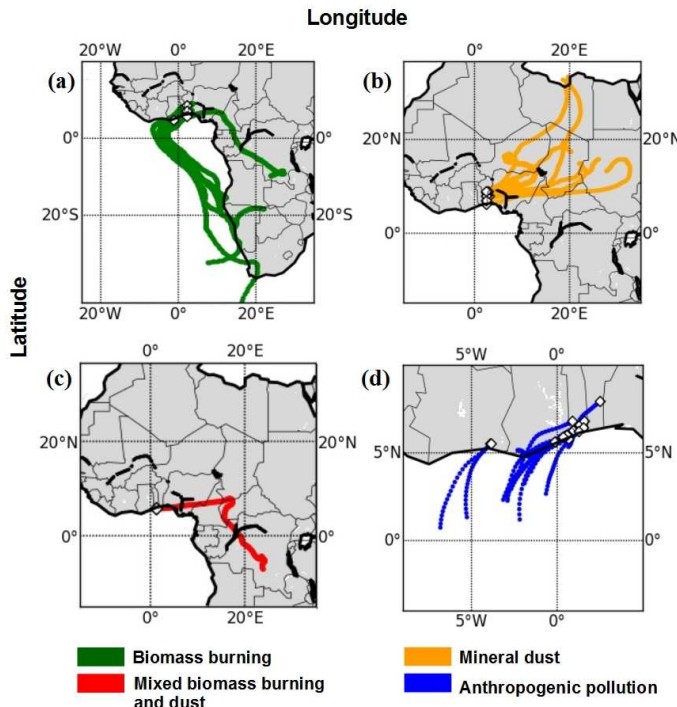

**Figure 2. Backward trajectories for the analyzed aerosol layers. Trajectories date**
**back 10 days for (a), 5 days for (b) and (c), and 1 day for (d).**

**3. Results**
**3.1. Aerosol vertical distribution**
Figure 3 shows a statistical analysis of $N_{fine}$, $N_{coarse}$ and $\sigma_{ext}$ derived from the *in-situ* measurements
of vertical profiles. The aerosol vertical structure is strongly related to the meteorological structure
of the atmosphere (see *Knippertz et al., 2017* for an overview of the DACCIWA field campaign*).*
Therefore wind vector and potential temperature profiles acquired with the aircraft have been
added to Figure 3 as a function of the dominating aerosol composition, introduced in Figure 1. The





data from individual vertical profiles were merged into 200 m vertical bins from the surface to 6
km amsl. The profiles were calculated using only individual profiles obtained outside of clouds.

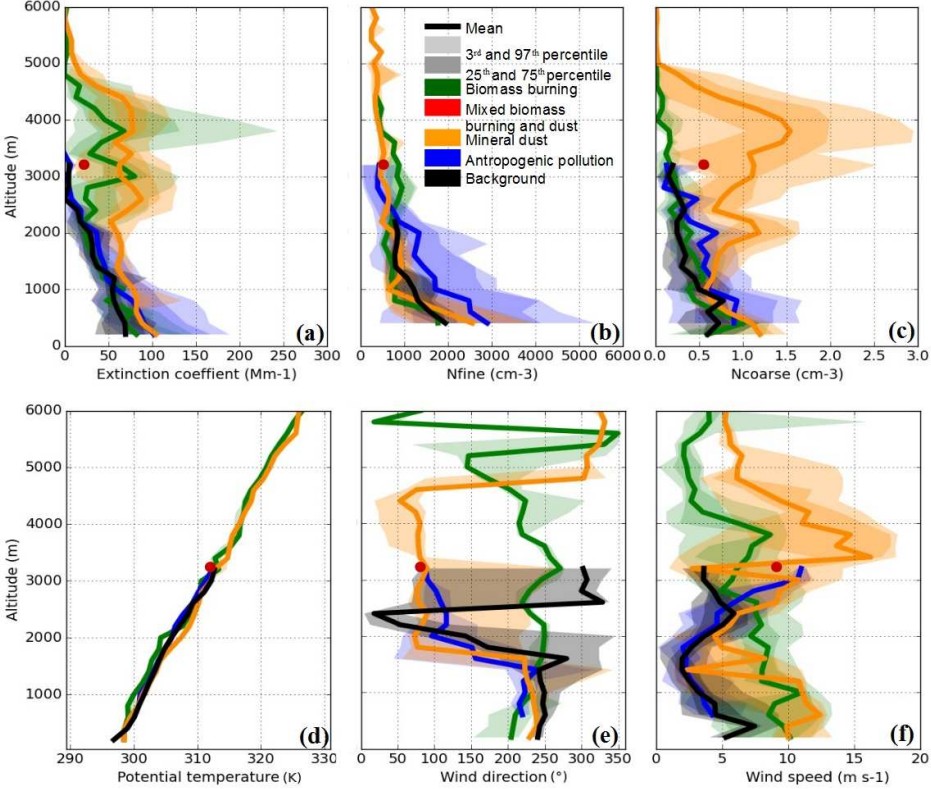

**Figure 3. Vertical layering of aerosols and meteorological variables for profiles for which aerosols dominated by biomass burning (green), dust (orange), mixed dust-biomass burning (red), anthropogenic pollution (blue) and background particles (black) were detected. The panels show profiles of (a) the extinction coefficient at 530 nm, (b) the particle number concentration in the range $0.005 < D_p < 1$ µm, (c) the particle number concentration in the range $1 < D_p < 5$ µm, (d) potential temperature, (e) the wind direction and (f) the wind speed. The colored areas represent the 3th, 25th, 75th and 97th percentiles of the data. The mixed dust-biomass burning plume is represented by a dot because it is derived from measurements during a SLR.**

The observed wind profiles highlight the presence of several distinct layers in the lower
troposphere. For cases related to dust, urban pollution and background condition, we clearly
observe the monsoon layer up to 1.5 km amsl which is characterized by weak to moderate wind
speeds (2 to 10 m s⁻¹, the later corresponding to dust cases) and a flow from the southwest (220-



250°). In all three air mass regimes, the monsoon layer is topped by a 500 to 700 m deep layer
characterised by a sharp wind direction change (from south-westerly to easterly). Weak wind
speeds (less than 5 m s$^{-1}$) are observed in urban pollution and background conditions, while higher
wind speeds are observed in the dust cases. Above 2.5 km amsl, the wind speed increases in the
urban pollution and dust cases and the wind remains easterly, indicating the presence of the
African easterly jet with its core typically farther north over the Sahel.. The maximum easterlies
are observed in the dust cases slightly below 3.5 km amsl (> 15 m s$^{-1}$). For the background cases,
the wind above the shear layer shifts to north-westerly and remains weak (~i.e. 5 m s$^{-1}$). Overall,
the wind profile associated with the biomass burning cases is quite different from the other three
cases, with a flow essentially from the south-southwest below 5 km amsl and higher wind speeds
in the lower 2 km amsl than above, and a secondary maximum of 7 m s$^{-1}$ at 4 km amsl.

The vertical distribution of aerosol particles was very inhomogeneous, both across separate
research flights and between individual plumes encountered during different periods of the same
flight. Measurements of aerosols within this analysis cover a broad geographic region, as shown in
Figure 1, which may explain some of the variability. SWA is subject to numerous anthropogenic
emission sources (e.g. road traffic, heavy industries, open agriculture fires, etc.) coupled to
biogenic emissions from the ocean and forests. These resulting large emissions are reflected in the
high variability of $\sigma_{ext}$, $N_{fine}$ and $N_{coarse}$ in the lower troposphere over SWA. Below 2.5 km amsl, $\sigma_{ext}$
showed a large heterogeneity with values ranging from 35 to 188 Mm$^{-1}$ between the 3$^{rd}$ and 97$^{th}$
percentile and a median value of 55 Mm$^{-1}$. The variability of $\sigma_{ext}$ values was slightly enhanced near
the surface and was correlated to $N_{fine}$ and $N_{coarse}$ which ranged from 443 to 5250 cm$^{-3}$ and from
0.15 to 1.6 cm$^{-3}$, respectively. Maximum surface $\sigma_{ext}$ was recorded in the anthropogenic pollution
plume of Accra where high $N_{fine}$ was sampled. The aerosol vertical profile is strongly modified
during biomass burning and dust events. The dust plume extends from 2 to 5 km amsl, and is
associated with transport from the dust sources in Chad and Sudan (see Figure 2) with the midlevel
easterly flow. The biomass burning plume is associated with transport from the southwest in a
layer of enhanced wind speed just below 4 km amsl as discussed above. Both layers showed
enhanced $\sigma_{ext}$ with median values of 68 Mm$^{-1}$ ($p_{03}$ = 12 Mm$^{-1}$; $p_{97}$ = 243 Mm$^{-1}$) in biomass burning
plumes and 78 Mm$^{-1}$ ($p_{03}$ = 45 Mm$^{-1}$; $p_{97}$ = 109 Mm$^{-1}$) in dust plumes. As expected, the extinction
profile was strongly correlated to $N_{fine}$ for biomass burning layers and $N_{coarse}$ for dust layers.

A prominent feature in the vertical profiles is the presence of fine particles up to 2.5 km amsl
outside of biomass burning or dust events. $\sigma_{ext}$, $N_{fine}$ and $N_{coarse}$ continuously decrease with altitude,





most likely due to vertical mixing of local emissions from the surface to higher levels. Therefore,
the regional transport of locally emitted aerosols was not limited to the surface but occurred also at
higher altitude. Recently, numerical tracer experiments performed for the DACCIWA airborne
campaign period have demonstrated that a combination of land–sea surface temperature gradients,
orography-forced circulation and the diurnal cycle of the wind along the coastline favor the
vertical dispersion of pollutants above the boundary layer during daytime *(Deroubaix et al., 2019;*
*Flamant et al., 2018a)*. Because of these complex atmospheric dynamics, aerosol layers transiting
over the Gulf of Guinea in the free troposphere could be contaminated by background or urban
pollution aerosols from the major coastal cities.

**3.2. Aerosol size distribution**
Figure 4a shows the range of variability of the number and volume size distributions measured
during DACCIWA. These are extracted from the SLRs identified in Fig.1. Figure 4b shows the
same composite distribution normalized by CO concentration in order to account for differences in
the amount of emissions from combustion sources.

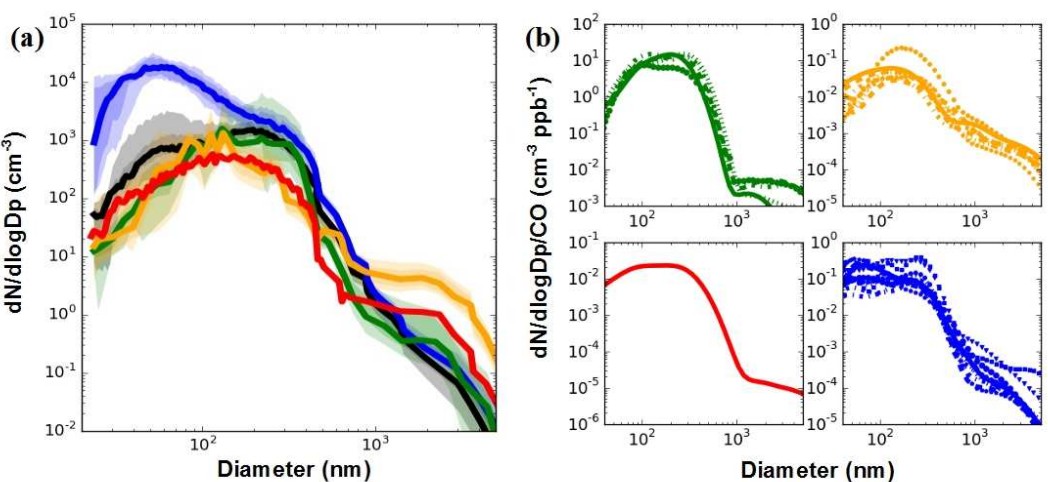

**Figure 4. (a) Statistical analysis of number size distributions with colored areas representing**
**the 3[th], 25[th], 75[th] and 97[th] percentiles of the data and (b) number size distributions normalized**
**to CO for plumes dominated by biomass burning (green), dust (orange), mixed dust-biomass**
**burning (red), anthropogenic pollution (blue) and background particles (black).**

Considerable variability in the number concentration of the size distributions, up to approximately
2 orders of magnitude, was observed for a large fraction of the measured size range. The size





distributions varied both for different aerosol types and for a given aerosol class. This reflects the relative wide range of different conditions that were observed over the region, both in terms of sources, aerosol loading, and lifetimes of the plumes.

In particular for ultrafine particles with diameters below 100 nm, large differences are observed, with an increase as large as a factor of 50 in urban plumes, which reflects concentration increase from freshly formed particles. Interestingly elevated number concentrations of these small-diameter particles were also observed in some dust layers. Comparing the particle size distribution of the different dust plumes sampled during the field campaign, a variation as large as a factor of 20 in the number concentration of ultrafine particles is found. Their contribution decreased with height as reflected by higher small particle number recorded in dust plumes below 2.5 km amsl (Figure 3b and 4b). As the composite urban size distributions showed a relatively similar ultrafine mode centered at 50 nm, dust layers have most likely significant contributions from anthropogenic pollution aerosol freshly emitted in SWA. The ultrafine mode was not observed in biomass burning size distributions, even though dust and biomass burning plumes were sampled in the same altitude range. We interpret this observation with dust plumes transported below 2.5 km amsl that were sampled over the region of Savè (8°01'N, 2°29'E; Benin) near the identified urban air mass transported northeastwards from Lomé and/or Accra and which may have collected significant fresh pollution on their way, whereas biomass burning plumes collected at the same altitude and sampled over Ivory Coast south of the Abidjan pollution plumes may not have been affected by significant direct pollution (Figure 1).

The accumulation mode was dominated by two modes centered at $D_{p,g}$ ~ 100 and 230 nm depending on the aerosol plume. The particle size distributions for biomass burning plumes were generally dominated by an accumulation mode centered at $D_{p,g}$ ~ 230 nm. Despite the relative wide range of sources and lifetimes of the biomass burning plumes sampled throughout the campaign (Figure 2), the $D_{p,g}$ in the accumulation mode showed little variation ($D_{p,g}$ from 210 to 270 nm) between the plumes. Similarly, previous field studies found accumulation mode mean diameters from 175 to 300 nm for aged biomass burning plumes, regardless of their age, transport time and source location *(Capes et al., 2008; Janhäll et al., 2010; Weinzierl et al., 2011; Sakamoto et al., 2015; Carrico et al. 2016)*. The coagulation rate can be very high in biomass burning plumes and can shape the size distribution over a few hours *(Sakamoto et al., 2016)*. It is worth noting that in the biomass burning and dust size distributions there is a persistent particle accumulation mode centered at ~100 nm that exceeds the amount of larger particles in some layers. This small mode is



420 unlikely to be related to long-range transport of biomass burning and Saharan dust emissions, as it
421 would be expected that particles in this size range would grow to larger particles through
422 coagulation relatively quickly. As similar concentrated accumulation modes of particles have been
423 observed in background plumes, it suggests the entrainment of background air from the boundary
424 layer in dust and biomass burning plumes. This is supported by remote sensing observations on 2
425 and 5 July 2016 *(Flamant et al., 2018a; Deroubaix et al., 2019).*

426

427 The number concentration of large super-micron particles was strongly enhanced in the mineral
428 dust layers. The peak number concentration displayed a broad shape at $D_{p,g} \sim 1.8$ µm, which is
429 comparable to literature values of other long-range transported dust aerosols *(Weinzierl et al.,*
430 *2011; Ryder et al., 2013; Denjean et al., 2016; Liu et al., 2018).* The relatively homogeneous $D_{p,g}$
431 in the coarse mode of dust ($D_{p,g}$ from 1.7 to 2.0 µm) suggests low internal mixing with other
432 atmospheric species. Besides, the volume size distribution in urban plumes showed significant
433 presence ($\sim 65\%$ of the total aerosol volume) of large particles with diameters of $\sim 1.5 - 2$ µm,
434 which were also observed in background conditions. This coarse mode has most likely significant
435 contributions from sea salt particles, as plumes arriving from the cities were transported at low
436 altitude over the ocean (Fig. 3).

437

438 **3.3. Aerosol optical properties**
439 *SSA* is one of the most relevant intensive optical properties because it describes the relative
440 strength of the aerosol scattering and absorption capacity and is a key input parameter in climate
441 models *(Solmon et al., 2008).* Figure 5 shows the spectral *SSA* for the different SLRs considered in
442 this study.

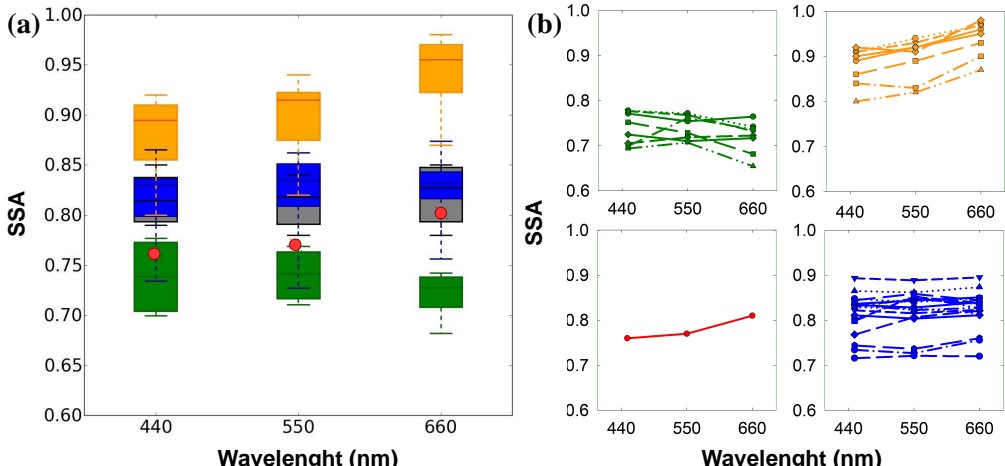

443

**Figure 5. (a) Statistical analysis of single scattering albedo at 450, 550 and 660 nm for**
**plumes dominated by biomass burning (green), dust (orange), mixed dust-biomass burning**
**(red), anthropogenic pollution (blue) and background particles (black). The boxes enclose the**
**25[th] and 75[th] percentiles, the whiskers represent the 5[th] and 95[th] percentiles and the**
**horizontal bar represents the median. (b) Spectral SSA for the different individual plumes**
**considered in this study. The mixed dust-biomass burning plume is represented by a dot**
**because it is derived from measurements during only one SLR.**

451

The highest absorption (lowest SSA) at all three wavelengths was observed for biomass burning

aerosols. SSA values ranged from 0.69–0.78 at 440 nm, 0.71–0.77 at 550 nm and 0.65–0.76 at 660

nm. This is on the low side of the range of values (0.73–0.93 at 550 nm) reported over West Africa

during DABEX for biomass burning plumes mixed with variable proportion of mineral dust

*(Johnson et al., 2008)*. No clear tendency was found for the spectral dependence of SSA, which in

some of the cases decreased with wavelength and in others were very similar to each other at all

three wavelengths. The strongest spectral dependence of SSA was observed for biomass burning

plumes with the lowest absorption (highest SSA) at 440 nm. Laboratory experiments have shown

that strongly absorbing biomass burning particles tend to have a weak wavelength-dependent

absorption, while weakly light-absorbing particles tend to have a strong wavelength dependent

absorption *(McMeeking et al., 2014; Zhai et al., 2017)*, which is consistent with results in this

study.

SSA values of anthropogenic pollution aerosols were generally intermediate in magnitude with

median values of 0.81 at 440 nm, 0.82 at 550 nm and 0.82 at 660 nm. Our data show that the value

of SSA varied significantly for the different plumes. Some pollution aerosols absorb almost as


strongly as biomass burning aerosols with SSA(550nm) values as low as 0.72, whereas the highest
SSA(550nm) value observed was 0.86. In addition, the absorption properties of urban aerosol
varied greatly between the sampled plumes for smoke of apparent same geographic origin. For
example, we measured SSA(550nm) values from 0.72 to 0.82 in the Accra pollution outflow. The
variability in SSA values may be due to the possible contribution of emissions from different cities
to the sampled pollution plumes (*Deroubaix et al., 2019)*, thus having different combustion
sources and chemical ages. Past in-situ measurements of aerosol optical properties over SWA cities
appear, unfortunately, to be absent from literature. However, the flat spectral dependence of SSA
appears to be anomalous for anthropogenic pollution aerosols, as SSA is expected to decrease with
increasing wavelength (*Dubovick et al., 2002; Di Biagio et al., 2016).* As during DACCIWA SSAs
of anthropogenic pollution aerosols reached similar values to those of background aerosols, it
suggests a large contribution of the latter to the aerosol optical properties of the mixture.
The magnitude of SSA increased at the three wavelengths when dust events occurred. It is
important to note that the measurements exclude a significant portion of the coarse-mode aerosol
due to poor inlet passing efficiency of larger aerosol particles (a 50% size cut around 5 µm), which
may result in an overestimate of SSA. Despite this limitation, our measurements are comparable to
one another and to previous *in-situ* measurements by taking into account the sampling inlet. Large
variations in SSA were obtained with values ranging from 0.76–0.92 at 440 nm, 0.81–0.94 at 550
nm and 0.81–0.97 at 660 nm. Compared with the literature for transported dust, lower values were
obtained in the present study for few cases. For example, *Chen et al. (2011)* reported SSA(550 nm)
values of 0.97±0.02 during NAMMA (a part of AMMA operated by NASA) using an inlet with a
comparable sampling efficiency. The lower values from DACCIWA reflect inherently more
absorbing aerosols in some dust plumes. In contrast to fire plumes, the SSA of dust aerosol showed
a clear increasing trend with wavelength. This behavior is likely due to the domination of large
particles in dust aerosol, which is in agreement to similar patterns observed in dust source regions
(*Dubovik et al., 2002).* Moreover, an increase of SSA is observed with wavelength for mixed dust-
smoke aerosol, suggesting that the aerosol particles were predominantly from dust, albeit mixed
with a significant loading of biomass burning.






| | | SSA(450) | SSA(550) | SSA(660) | MEC(450) | MEC(550) | MEC(660) | g(450) | g(550) | g(660) | SAE |
|---|---|---|---|---|---|---|---|---|---|---|---|
| **Mineral dust** | median | 0.88 | 0.90 | 0.93 | 0.74 | 0.68 | 0.66 | 0.74 | 0.72 | 0.69 | -0.35 |
| | 3[th] | 0.82 | 0.82 | 0.86 | 0.38 | 0.38 | 0.39 | 0.69 | 0.67 | 0.65 | -0.56 |
| | 25[th] | 0.85 | 0.87 | 0.90 | 0.43 | 0.43 | 0.43 | 0.73 | 0.72 | 0.67 | -0.48 |
| | 75[th] | 0.91 | 0.93 | 0.96 | 0.94 | 0.85 | 0.85 | 0.75 | 0.74 | 0.72 | -0.25 |
| | 97[th] | 0.92 | 0.95 | 0.97 | 1.57 | 1.37 | 1.21 | 0.78 | 0.76 | 0.72 | -0.12 |
| **Biomass burning** | median | 0.74 | 0.76 | 0.72 | 1.91 | 1.62 | 1.34 | 0.69 | 0.68 | 0.61 | 1.07 |
| | 3[th] | 0.70 | 0.72 | 0.66 | 0.94 | 1.45 | 1.22 | 0.64 | 0.65 | 0.59 | 0.59 |
| | 25[th] | 0.70 | 0.76 | 0.71 | 1.67 | 1.48 | 1.27 | 0.69 | 0.65 | 0.60 | 0.83 |
| | 75[th] | 0.77 | 0.77 | 0.74 | 1.86 | 1.65 | 1.55 | 0.72 | 0.68 | 0.62 | 1.15 |
| | 97[th] | 0.78 | 0.77 | 0.76 | 2.38 | 1.92 | 1.58 | 0.73 | 0.68 | 0.63 | 1.64 |
| **Mixed dust-Biomass burning** | median | 0.76 | 0.77 | 0.81 | 1.58 | 1.40 | 1.30 | 0.73 | 0.66 | 0.64 | 0.38 |
| | 3[th] | - | - | - | - | - | - | - | - | - | - |
| | 25[th] | - | - | - | - | - | - | - | - | - | - |
| | 75[th] | - | - | - | - | - | - | - | - | - | - |
| | 97[th] | - | - | - | - | - | - | - | - | - | - |
| **Anthropogenic Pollution** | median | 0.83 | 0.84 | 0.85 | 2.60 | 2.49 | 1.90 | 0.60 | 0.61 | 0.62 | 0.75 |
| | 3[th] | 0.78 | 0.79 | 0.81 | 0.70 | 1.24 | 0.54 | 0.60 | 0.59 | 0.54 | 0.30 |
| | 25[th] | 0.80 | 0.82 | 0.83 | 2.14 | 2.25 | 1.53 | 0.62 | 0.60 | 0.56 | 0.65 |
| | 75[th] | 0.84 | 0.86 | 0.85 | 3.51 | 2.96 | 2.53 | 0.69 | 0.62 | 0.67 | 0.89 |
| | 97[th] | 0.87 | 0.88 | 0.90 | 3.70 | 4.83 | 2.74 | 0.73 | 0.64 | 0.70 | 0.94 |

**Table 2. Single scattering albedo, extinction mass efficiency, asymmetry parameter, single**
**scattering albedo and scattering Ångstrom exponent for the dominant aerosol classification.**
As shown in Table 2, the observed variability of *SSA* reflects a large variability for *MEC* at 550nm,
which spans a wide range from 0.38 to 1.37 m$^2$ g$^{-1}$, 1.45 to 1.92 m$^2$ g$^{-1}$ and 1.24 to 4.83 m$^2$ g$^{-1}$ for
dust, biomass burning for anthropogenic polluted aerosols, respectively. According to Mie theory,
*MEC* is heavily influenced by the mass concentrations in the accumulation mode where the aerosol
is optically more efficient in extinguishing radiation. We found *MEC* to be positively correlated
with *SAE* (not shown) with measurement wavelengths (450–660 nm), which agrees with Mie
theory. In contrast, the values of *g* appear to differ only little between the sampled plumes for a
given aerosol class. We found *g* in the range of 0.67–0.76 for dust, 0.65–0.68 for biomass burning
and 0.59–0.64 for anthropogenic polluted aerosols at 550 nm. *g* values in dust plumes were high,
which is expected due to the presence of coarse particles contributing to forward scattering.

This analysis includes sampled aerosols originating from different source regions and having
undergone different aging and mixing processes, which could explain some of the variability. The



impact of these factors on the magnitude and spectral dependence of optical parameters will be
investigated in the following section.
**4. Discussion**
**4.1. Contribution of local anthropogenic pollution on aerosol absorption properties**
Figure 6 shows the vertical distribution of *SSA, SAE* and *NOx* mixing ratio for the dominant
aerosol classification. In dust plumes, if we exclude the case of mixing with biomass burning
aerosol, SSAs were fairly constant above 2.5 km amsl with values ranging between 0.90 and 0.93
at 550 nm, in agreement with values reported over dust source regions *(Schladitz et al., 2009;*
*Formenti et al., 2011; Ryder et al., 2013, 2018).* Despite the range of sources identified during
DACCIWA, dust absorption properties do not seem to be clearly linked to particle origin or time
of transport. Aerosols were more absorbing within the low-altitude dust plumes with SSA values
dropping to 0.81. *SAE* values exhibited simultaneously a sharp increase close to zero below 2.5 km
amsl. This is consistent with a higher concentration of fine particles, though the value of *SAE* was
still much lower than for pollution or background aerosol (i.e. where it is typically $> 0.2$), which
means that scattering was still dominated by larger particles. The decrease in NOx with height
further indicates the concurrent influence by emissions from pollution sources in the low-altitude
dust plumes. Based on these observations, the strong variation in the light-absorption properties of
dust-dominated aerosol over SWA could be attributed to the degree of mixing into the vertical
column with either freshly emitted aerosols from urban/industrial sources or long-range
transported biomass burning aerosol.
One of the critical factors in the calculation of aerosol direct and semi-direct radiative effects is the
mixing state of the aerosols, which can significantly affect absorbing properties. There were no
direct observational constraints available on this property during the DACCIWA airborne
campaign. However, we investigated the probable aerosol mixing state by calculating composite
SSA from the aerosol size distribution. On the basis of Figure 4, dust size distribution showed only
minor discrepancies in the mean and standard deviation of the coarse mode but significant
differences in the balance between fine and coarse modes, which suggests low internal mixing of
dust with other atmospheric species. The size distributions of mixed dust-pollution have been
deconvoluted by weighting the size distributions of mineral dust and anthropogenic pollution
aerosol averaged over the respective flights. This assumes that dust was externally mixed with the
anthropogenic pollution particles and assumes a homogeneous size distribution for the dust and
anthropogenic pollution aerosol throughout a flight. $\sigma_{scat}$ and $\sigma_{abs}$ were then calculated using Mie





theory from each composite size distributions and the corresponding *k* and *m*. The resulting $\sigma_{scat}$
and $\sigma_{abs}$ were used to calculate a composite *SSA*. A similar calculation was performed for the mixed
dust-biomass burning case. Figure 6 shows a good agreement with the observations of *SSA*,
implying that external mixing appears to be a reasonable assumption to compute aerosol direct and
semi-direct radiative effects in these dust layers for modeling applications. This is consistent with
the filter analysis performed during AMMA and SAMUM-2, which did not reveal any evidence of
internal mixing in both mixed dust-biomass burning and dust-anthropogenic pollution layers
*(Chou et al., 2008; Lieke et al., 2011; Petzold et al., 2011).*

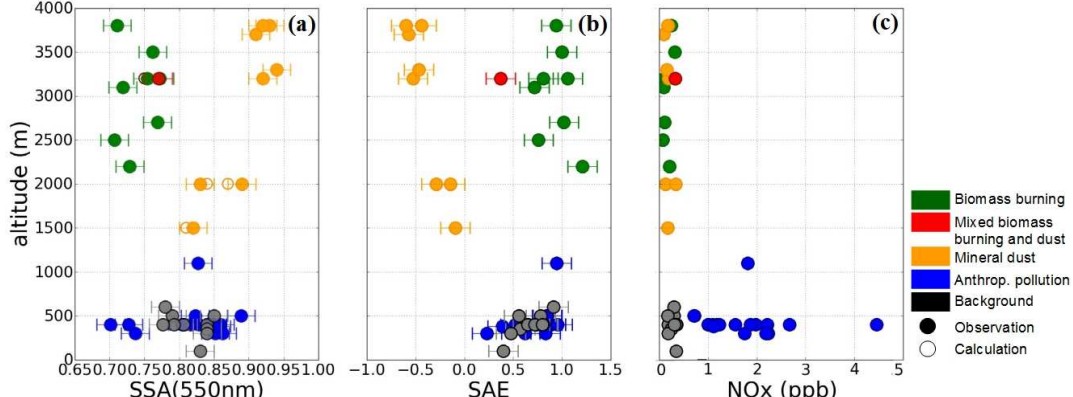

**Figure 6. Vertical distribution of (a) the single scattering albedo at 550 nm, (b) the scattering**
**Ångstrom exponent and (c) NOx mixing ratio for the dominant aerosol classification. In**
**panel (a), full circles represent *SSA* measurements and empty circles represent composite**
***SSA* calculated by deconvoluting size distribution measurements in mixed dust layers and**
**assuming an external mixing state.**

Figure 6 indicates markedly different processes affecting optical properties of biomass burning
aerosols. *SSA*, *SAE* and NOx of biomass burning plumes did not significantly vary with height
from 2.2 to 3.8 km amsl. Moreover, the size distribution of biomass burning aerosols for the
observed cases did not show significant contribution of ultrafine particles (Figure 4). These
observations seem to indicate that the absorption properties of biomass burning plumes were not
affected by direct pollution emissions, probably due to the remote location of the sampled biomass
burning plumes as discussed in section 3.2. The optical properties of aerosols are determined by
either the aerosol chemical composition, the aerosol size distribution, or both. Changes in the size
distribution of biomass burning aerosol due to coagulation and condensation have been shown to
alter the *SSA*, as particles increase towards sizes for which scattering is more efficient *(Laing et al.,*



*2016)*. Variations in particle chemical composition, caused by source emissions and aging
processes associated with gas-to-particle transformation and internal mixing, has been shown to
change the *SSA (Abel et al., 2003; Petzold et al., 2011)*. The contributions from size distribution
and chemical composition to the variation of *SSA* in biomass burning plumes will be explored in
section 4.2.

In the boundary layer, the similar *SSA* and *SAE* in anthropogenic pollution and background plumes
suggests that background aerosol may be rather called background pollution originating from a
regional background source in the far field. Our analysis of the spectral dependence of *SSA*
showed no apparent signature of anthropogenic pollution aerosols (see section 3.3) despite a strong
increase of aerosol number concentrations in air masses crossing urban centers (see section 3.2).
This can be explained by two factors: First, the majority of accumulation mode particles were
present in the background, while the large proportion of aerosols emitted from cities resided in the
ultrafine mode particles that have less scattering efficiencies (Figure 4). Second, large amounts of
absorbing aerosols in the background can minimize the impact of further increase of absorbing
particles to the aerosol load. We did not find any correlation between the values of *SSA* and their
spectral dependence, which suggests that the variability in *SSA* cannot be attributed to different
contributions of marine aerosol in pollution plumes. The high CO values (~180 ppb) observed in
background conditions further indicates a strong contribution of combustion emissions at the
surface. Recent studies showed a large background of biomass burning transported from the
Southern Hemisphere in SWA that dominated the aerosol chemical composition in the boundary
layer *(Menut et al., 2018; Haslett et al., 2019)*. The high absorbing properties (*SSA*~0.81 at
550nm) of background aerosols is consistent with being a mixture of aged biomass burning and
Atlantic marine aerosol. Moreover *SSA* of background aerosol was lower than previously reported
over the Southern Atlantic (Ascension Island) outside the fire season in Central Africa *(Zuidema*
*al., 2018)*, which supports this conclusion. These results highlight that aerosol optical properties at
the surface were dominated by the widespread biomass burning particles at regional scale.

**4.2. Aging as a driver for absorption enhancement of biomass burning aerosol**
In order to determine the contributions from size distribution and chemical composition to the
variation of *SSA* in biomass burning plumes, *SSA* is presented as a function of *SAE* and *k* in Figure
7a and b, respectively. *k* was iteratively varied to reproduce the experimental scattering and
absorption coefficients, as described in section 2.3.1. It appears that the variation of the size
distribution (assessed via *SAE* in Figure 7a) had minimal impact in determining the variability of





*SSA*. Thus, the observations suggest that there was no effect of plume age on the size distribution,
consistent with previous observations of size distribution in aged biomass burning plumes
*(Sakamoto et al., 2015; Carrico et al., 2016; Laing et al., 2016)*. Using a Lagrangian
microphysical model, *Sakamoto et al. (2015)* have shown a rapid shift to larger sizes for biomass
burning plumes within the first hours of aging. Less drastic but similarly rapid growth by
coagulation was seen by *Capes et al. (2008)* in their box model. Given that the biomass burning
plumes sampled during DACCIWA had more than 5 days in age, the quick size-distribution
evolution within the early plume stages might explain the limited impact of the size distribution on
the *SSA*.

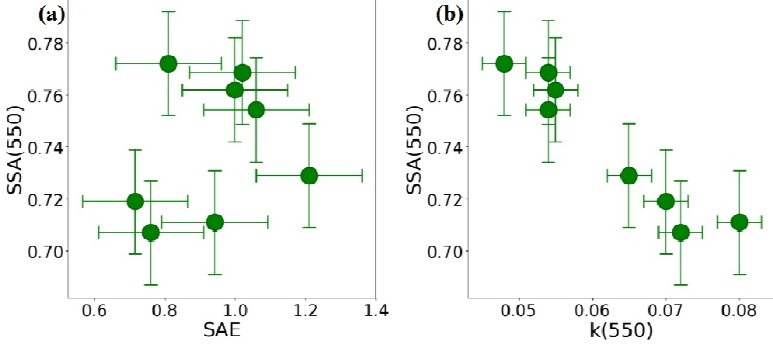

**Figure 7. Contribution to single scattering albedo (a) from particle size (assessed via *SAE*)**
**and (b) from composition (assessed via *k*) in biomass burning plumes.**

In contrast, Figure 7b shows that *SSA* variability was strongly influenced by the variability in
composition of biomass burning aerosol. Although there is some variability between the results
from different plumes, overall there was a consistent decrease in *SSA* with increasing *k*, implying a
high contribution for light-absorbing particles. The observed variability of *SSA* is reflected in a
large variability of *k*, which is estimated to span in the large range 0.048–0.080 at 550 nm. No
clear tendency was found for the wavelength dependence of *k*, which in some of the cases
increases with wavelength and in others decreases (not shown). Correspondingly, values of *AAE* in
biomass burning plumes ranged from 0.9 to 1.1 with a median value of 1.0. Theoretically, fine-
mode aerosol with absorption determined exclusively by BC would have *AAE* equal to 1.0, since
BC is expected to have a spectrally constant *k (Bond et al., 2013)*. Therefore, the low *SSA* values
observed in biomass burning plumes over SWA and the small spectral variation of *k* both suggest





that BC is the dominant absorber in the visible and near-IR wavelengths for these biomass burning
aerosols.

Compared with past in-situ measurements of aged biomass burning aerosol, *SSA* values over SWA
(0.71–0.77 at 550 nm) are at the lower end of those reported worldwide (0.73–0.99 at 550 nm)
*(Maggi et al., 2003; Reid et al. 2005; Johnson et al., 2008; Corr et al. 2012; Laing et al. 2016)*.
This can be attributed in part to the high flaming versus smoldering conditions of African smoke
producing more BC particles *(Andreae and Merlet, 2001; Reid et al., 2005)*, which inherently have
low *SSA* compared to other regions *(Dubovick et al.* 2002). However, *SSA* values over SWA are
significantly lower than the range reported near emission sources in sub-Saharan Africa and over
the southeast Atlantic, where values span over 0.84–0.90 at 550 nm *(Haywood et al., 2003b;*
*Pistone et al., 2019)*. Recent observations carried out on Ascension Island to the south-west of the
DACCIWA region showed that smoke transported from Central and South African fires can be
very light absorbing over the July-November burning season but *SSA* values were still higher
(0.80±0.02 at 530 nm; *Zuidema et al., 2018*) than those reported over SWA. A possible cause of
the lower *SSA* in SWA is that Ascension Island is much closer to the local sources and the aerosol
is therefore less aged.

Currently there are few field measurements of well-aged biomass burning emissions. Our
knowledge of biomass burning aerosol primarily comes from laboratory experiments and near-
field measurements taken within a few hours of a wildfire *(Abel et al., 2003; Yokelson et al., 2009;*
*Adler et al., 2011; Haywood et al., 2003b; Vakkari et al., 2014; Zhong and Jang, 2014; Forrister*
*et al., 2015; Laing et al., 2016; Zuidema et al., 2018)*. Exception made of the study by *Zuidema et*
*al. (2018)* over the southeast Atlantic, it is generally found that the aged biomass burning aerosol
particles are less absorbing than freshly emitted aerosols due to a combination of condensation of
secondary organic species and an additional increase in size by coagulation. This is in contrasts to
our results showing that *SSA* of biomass burning aerosols were significantly lower than directly
after emission and that the evolution of *SSA* occurred long time after emission.

There are three possible explanations for these results. First, one must consider sample bias. As
regional smoke ages, it can be enriched by smoke from other fires that can smolder for days
producing large quantities of non-absorbing particles, thereby increasing the mean *SSA (Reid et al.,*
*2005; Laing et al., 2016)*. However, during DACCIWA, biomass burning plumes were transported
over the Atlantic Ocean and were probably less influenced by multiple fire emissions. Second,



there is evidence that fresh BC particles become coated with sulfate and organic species as the
plume ages in a manner that enhances their light absorption *(Lack et al., 2012; Schwarz et al.,*
*2008)*. Finally, organic particles produced during the combustion phase can be lost during the
transport through photobleaching, volatilization and/or cloud-phase reactions *(Clarke et al., 2007;*
*Lewis et al., 2008; Forrister et al., 2015)*, which is consistent with the low *SSA* and *AAE* values we
observed. Assessing whether these aging processes impact the chemical components and
henceforth optical properties of transported biomass burning aerosol would need extensive
investigation of aerosol chemical composition that will be carried out in a subsequent paper.

**5. Conclusions**

This paper provides an overview of *in-situ* airborne measurements of vertically resolved aerosol
optical properties carried out over SWA during the DACCIWA field campaign in June-July 2016.
The peculiar dynamics of the region lead to a chemically complex situation, which enabled
sampling various air masses, including long-range transport of biomass burning from Central
Africa and dust from Sahelian and Saharan sources, local anthropogenic plumes from the major
coastal cities, and mixtures of these different plumes. This work fills a research gap by providing,
firstly, key climate relevant aerosol properties *(SSA, MEC, g, SAE, AAE)* and secondly,
observations of the impact of aging and mixing processes on aerosols optical properties.

The aerosol vertical structure was very variable and mostly influenced by the origin of air mass
trajectories. While aerosol extinction coefficients generally decreased with height, there were
distinct patterns of profiles during dust and biomass burning transport to SWA. When present,
enhanced values of extinction coefficient up to 240 Mm$^{-1}$ were observed in the 2–5 km amsl range.
These elevated aerosol layers were dominated by either dust or biomass burning aerosols, which is
consistent with what would be expected on the basis of the atmospheric circulations during the
monsoon season *(McConnell et al., 2008; Knippertz et al., 2017)*. However, during one flight a
mixture of dust and biomass burning was found in a layer at around 3 km amsl, implying that there
may be substantial variability in the idealized picture. In the lower troposphere, the large
anthropogenic pollution plumes extended as far as hundreds of kilometers from the cities emission
sources and were not limited to the boundary layer but occurred also at higher levels up to 2.5 km
amsl, which is explained by vertical transport and mixing processes, partly triggered by the
orography of SWA *(Deroubaix et al., 2019; Flamant et al., 2018a)*. The analysis of the aerosol size
distributions, *SAE* and NOx suggests a strong mixing of dust with anthropogenic pollution
particles in dust layers transported below 2.5 km amsl, whereas biomass burning plumes that were


transported more northward were not affected by this mixing. Both transport pathways and vertical
structures of biomass burning and dust plumes over SWA appear to be the main factors affecting
the mixing of anthropogenic pollution with dust and biomass burning particles.

The aerosol light absorption in dust plumes was strongly enhanced as the result of this mixing. We
find a decrease of *SSA(550nm)* from 0.92 to 0.81 for dust affected by anthropogenic pollution
mixing compared to the situation in which the dust plumes moved at higher altitudes across SWA.
Comparison of the particle size distributions of the different dust plumes showed a large
contribution of externally mixed fine mode particles in mixed layers, while there was no evidence
for internal mixing of coarse particles. Concurrent optical calculations by deconvoluting size
distribution measurements in mixed layers and assuming an external mixing state allowed to
reproduce the observed *SSA*s. This implies that an external mixing would be a reasonable
assumption to compute aerosol direct and semi-direct radiative effects in mixed dust layers.

Despite a strong increase of aerosol number concentration in air masses crossing urban
conglomerations, the magnitude of the spectral *SSA*s was comparable to the background.
Enhancements of light absorption properties were seen in some pollution plumes, but were not
statistically significant. A persistent spectral signature of biomass burning aerosols in both
background and pollution plumes highlights that the aerosol optical properties in the boundary
layer were strongly affected by the ubiquitous biomass burning aerosols transported from Central
Africa *(Menut et al., 2018; Haslett et al., 2019)*. The large proportion of aerosols emitted from
cities that resided in the ultrafine mode particles have limited impact on already elevated amounts
of accumulation mode particles having a maximal absorption efficiency. As a result, in the
boundary layer, the contribution from local city emissions to aerosol optical properties were of
secondary importance at regional scale compared with this large absorbing aerosol mass. While
local anthropogenic emissions are expected to rise as SWA is currently experiencing major
economic and population growth, there is increasing evidence that climate change is increasing the
frequency and distribution of fire events *(Joly et al., 2015)*. In terms of future climate scenarios
and accompanying aerosol radiative forcing, whether the large biomass burning events that occur
during the monsoon season would limit the radiative impact of increasing anthropogenic
emissions, remains an open and important question.



The *SSA* values of biomass burning aerosols transported in the free troposphere were very low
(0.71–0.77 at 550 nm) and have only rarely been observed in the atmosphere. The variability in
*SSA* was mainly controlled by the variability in aerosol composition (assessed via *k*) rather than by
variations in the aerosol size distribution. Correspondingly, values of *AAE* ranged from 0.9 to 1.1,
suggesting that BC particles were the dominant absorber in the visible for these biomass burning
aerosols. In recent years the southern Atlantic Ocean, especially the area of the west coast of
Africa, became an increasing focus in the research community, through the ORACLES/LASIC
(ObseRvations of Aerosols above CLouds and their intEractionS/Layered Atlantic Smoke
Interactions with Clouds), AEROCLO-sA (AErosol RadiatiOn and CLOuds in Southern Africa –
AEROCLO-SA) and CLARIFY (Cloud and Aerosols Radiative Impact and Forcing) projects
*(Zuidema et al.; 2016; Zuidema et al.; 2018; Formenti et al.; 2019)*. Comparison with literature
showed a consistent picture of increasing absorption enhancement of biomass burning aerosol
from emission to remote locations. Further, the range of *SSA* values over SWA was slightly lower
than that reported on Ascension Island to the south-west of the DACCIWA region, which
underscores that the evolution of *SSA* occurred long time after emission. While the mechanism
responsible for this phenomenon warrants further study, our results support the growing body of
evidence that the optical parameters used in regional/global climate modeling studies, especially
absorption by biomass burning aerosols, have to be better constrained using these recent
observations to determine the direct and semi-direct radiative effects of smoke particles over this
region *(Mallet et al. 2019)*. In particular and regarding the very high absorbing properties of
smoke, specific attention should be dedicated to the semi-direct effect of biomass burning aerosols
at the regional scale and its relative contribution to the indirect radiative effect.

We believe the set of DACCIWA observations presented here is representative of the regional
mean and variability in aerosol optical properties that can be observed during the monsoon season
over SWA, as the main dynamical features were in line with climatology *(Knippertz et al., 2017)*.
This is why results from the present study will serve as input and constraints for climate modeling
to better understand the impact of aerosol particles on the radiative balance and cloud properties
over this region and also will substantially support remote sensing retrievals.



*Data availability.*
All data used in this study are publicly available on the AERIS Data and Service Center, which can
be found at http://baobab.sedoo.fr/DACCIWA.

*Author contributions.*
CD conducted the analysis of the data and wrote the paper. CD, TB, FB, NM, AC, PD, JB, RD, KS
and AS operated aircraft instruments and processed and/or quality-controlled data. MM provided
expertise on aerosol-climate interaction processes. CF and PK were PIs, who led the funding
application and coordinated the DACCIWA field campaign. All co-authors contributed to the
writing of the paper.

*Acknowledgements.*
The research leading to these results has received funding from the European Union 7th
Framework Programme (FP7/2007-2013) under Grant Agreement no. 603502 (EU project
DACCIWA: Dynamics-aerosol-chemistry-cloud interactions in West Africa). The European
Facility for Airborne Research (EUFAR, http://www.eufar.net/) also supported the project through
the funding of the Transnational Activity project OLACTA and MICWA. We thank the Service des
Avions Français Instrumentés pour la Recherche en Environnement (SAFIRE, a joint entity of
CNRS, Météo-France, and CNES) and operator of the ATR-42 for their support during the aircraft
campaign. Cyrielle Denjean thanks CNES for financial support. The authors would like to thank
Bruno Piguet (CNRM) and Michel Ramonet (LSCE) for their support in the data processing.



**Appendix 1. Summary of flight information. All flights were conducted during 2016.**

| Flight number | Date | Take off time (UTC) | Landing time (UTC) | Events observed |
|---|---|---|---|---|
| F17 | 29 June | 14:17 | 17:10 | Export of pollution from Lomé |
| F18 | 30 June | 12:52 | 16:29 | Export of pollution from Lomé |
| F19 | 1 July | 10:35 | 14:06 | Export of pollution from Accra |
| F20 | 2 July | 09:53 | 13:21 | Export of pollution from Lomé Dust outbreak |
| F21 | 2 July | 15:04 | 18:29 | Export of pollution from Lomé Biomass burning outbreak Mixed dust-biomass burning outbreak |
| F22 | 3 July | 09:54 | 13:29 | Export of pollution from Lomé |
| F24 | 6 July | 07:17 | 11:03 | Export of pollution from Abidjan |
| F27 | 8 July | 05:52 | 09:28 | Export of pollution from Accra |
| F28 | 8 July | 10:53 | 14:22 | Dust outbreak |
| F29 | 10 July | 10:31 | 14:11 | Export of pollution from Lomé Dust outbreak |
| F30 | 11 July | 07:19 | 11:01 | Export of pollution from Abidjan Biomass burning outbreak |
| F31 | 11 July | 13:48 | 16:42 | Biomass burning outbreak |
| F32 | 12 July | 13:56 | 17:20 | Export of pollution from Accra |
| F33 | 13 July | 12:40 | 16:11 | Biomass burning outbreak |
| F35 | 15 July | 09:32 | 13:00 | Export of pollution from Lomé |



**Appendix 2. Data inversion procedure to calculate the aerosol microphysical and optical**
**parameters.**

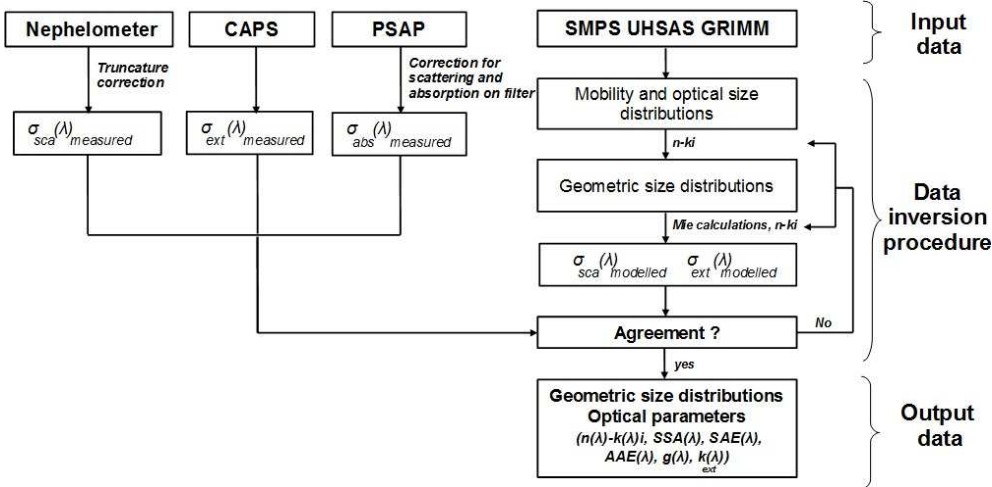




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
