# Peer review of "Light absorption properties of aerosols over Southern West Africa"

_Atmospheric Chemistry and Physics, 2019_

## Referee Comment (RC1) · Anonymous Referee #1 · 27 Sep 2019

General Comments

The authors present an overview of in-situ measurements of aerosol optical properties made during the DACCIWA aircraft measurement campaign over the coastal region of Southern West Africa in summer 2016. The region has a varied aerosol environment, in which they sampled mineral dust aerosol, biomass burning aerosol, and anthropogenic pollution. Along with the meteorological situation, instruments on the aircraft measured the aerosol size and vertical distributions, and the aerosol scattering, absorption and extinction coefficients. From this information the authors further derived the aerosol scattering and absorption Ångström exponents, the complex refractive index, the single scattering albedo (SSA), the mass extinction efficiency, and the asymmetry parameter. Using this information, the authors describe the vertical distribution of the aerosols with

respect to their optical properties, as well as showing that the SSA of the measured biomass burning aerosols is primarily a function of the imaginary part of the refractive index rather than of the size distribution. The derived SSA values of the biomass burning aerosols appear to be particularly low compared to other measurements from other regions of the world.

This is a very straightforward paper, and I appreciate that the scope of this work is primarily concerned with presenting the measurements and providing an interpretation of their significance. Within this scope, I would say that this is a successful paper, and hence I only have a few minor comments to make.

Specific comments

Figure 1: I assume that the black lines are the flight tracks in background conditions?

Section 2.2: as a qualitative comparison, I wonder if it would help to include one of these satellite images (MODIS or SEVIRI) over SWA with one of the flight tracks? This may be useful context to add and may aid the reader's understanding of the regional environment. Maybe such a figure may be more appropriate with respect to Section 2.3.2, and it may also be useful to have an image for each of the aerosol types.

Section 2.3.2: might it also be helpful to tabulate (briefly) the aerosol classification specifications? This may be helpful for quick reference when the reader wants to remind themselves of how the aerosols are classified in the later sections and figures.

Figure 5, labels: "wavelength"

---

## Referee Comment (RC2) · Anonymous Referee #2 · 11 Oct 2019

This paper describes the aerosol physical and optical properties observed during the DACCIWA campaign. The paper is based on in-situ measurements performed a-board the ATR-42. The paper is well written and the topic fits the ACP's scope. However, I have many remarks and some needs to be taken into account before this paper could be published in ACP.

Major comments

1. The title of this manuscript. I believe the title refers to the BB aerosols that were observed with really low SSA in comparison to all previous results. However, this part is described only on the 23th page. I would strongly recommend changing it to "aerosol optical properties overview during the DACCIWA campaign" or something similar.

[Figure]

2. First major comments about the Mie code used for this study. So you stated few lines (P8 L237-245) about how you used the Mie code to retrieve the refractive index of aerosols (reverse method) based on their physical and optical properties. First of all you are using a Mie code for dust and BB aerosols. In fact you measure the asymmetry factor (which is not that good with the TSI nephelometer) and found out that the particles are far from spherical. You never stated the possible issues with it. Then you never say a word for the mixing state you used. I found a word about it later when you say that the mixing was external which was consistent with previous observations. Have you tested the case with internal mixing (either core shell or just coating) ? Moreover, what is the acceptable difference between measurements and calculation to stop the iterative process (5%, 10% more ??? )? Is it just iterative process with an a priori k and n or it an optimal estimation method that will avoid any local minimum? Later on (P20-21, L548-556), you used the Mie code again but this time I believe not in a reverse mode. So you entered a k and n along with the size distribution and you get optical properties that you used to calculate a SSA. Which k and did you used? In Africa, there are many sources of dust and BB aerosols associated with many different k and n . . . How these calculations made with a Mie code, for spherical particles, prove that your aerosols are externally mixed ? You need here to prove it with a spheroidal code, for different mixing state.

3. Number concentration of coarse particles. The authors are showing number concentrations of particles from 1-5um. These concentrations are measured from the OPC-GRIMM. The concentrations are between 0 and 2.5 cm-3. Could you provide the reader the noise of this instrument regarding these large particles. Is it significant ?

4. During the field campaign, the ATR-42 performed vertical profiles at the beginning of each flight. These are used in Figure 3. On Figure 1, you highlighted the BB, dust and pollution plumes. How far are the vertical profiles from the plume observed ? I wonder how the variability of the vertical profile average is due to the distance to the actual plume. I had to wait until Figure 6 (P21) to have information about altitudes

at which you observed the different plumes. At P14 L366-370, P15 L398-399, you are showing that aerosols, within a layer located below 2.5km, are a efficiently mixed with anthropogenic emissions. Then on Figure 6 you are trying to interpret the optical variations throughout the vertical profile. From my understanding, if you are interested in "pure" BB or Dust particles than you should only show/interpret vertical profiles from 2.5km up to 4km.

Minor remarks

Page 2 L37 : I think something is missing to introduce properly the BB aerosols or just to skip a line here.

Page 2 L68 : It's usually called MEE that stands for Mass Extinction Efficiency and its units is m2/g not g/cm3. In the table2's caption you even called it extinction mass efficiency. Please be consistent throughout the manuscript. By the way how did you measure the mass concentration of aerosols ? Did you assume a density for each type ? And did you assume the same density for each mode ?

P3 L73-75 : Why do you introduce remote sensing data that you will never ever talk again ?

P3 L112 remove the 's' at aerosol

P 5 When DACCIWA took place (month, year) ? What season was it (apparently during the monsoon period)?

P6 L 176 : the CPC MARIE model needs to be described here. There is no paper associated with this instrument and I was unable to find any specifications on internet. Is it working with a flow regulator or a critical orifice? How is it counting in comparison to any TSI CPC ? You used the CPC measurements to infer the Nfine but you never say anything about a comparison, in term of number concentration, with the OPC, UHSAS or the SMPS. As this instrument is not (yet?) considered as a reference, you cannot use it like a reference. You could use the calibration data for the size distribution (PSL

and oil) to prove that this instrument was measuring the same number concentrations than the OPC.

P6 L 187 Replace 525 by 550nm.

P10 section 2.3.2 : A figure showing the AAE versus the SAE would have been appreciated here.

P10 L282 : remove the 's' at the end of theses

P11 L 301 : I believe that the background is also without the detectable influence of Dust and BB as well as anthropogenic emissions.

P13 L334 : The AEJ is a synoptic flow. It is always present over West Africa at a specific location (between 10-20°N between 2-6km). The ATR flew between 3-11°N. Therefore, I don't see how you would have been able to observe it and why you would have been able to observe it for specific cases and not for the others.

P13 L 335 : there are two dots after Sahel.

P13 L356-357 : According to the Nfine vertical profile, the BB plume is observed from 2.5 – 4km. Your sentence isn't clear for me. Is this between the surface up to 4km or as suggested by the Nfine vertical profile ?

Figure 4 : I think you could do something in this figure to show the altitude of each SLR. You never mentioned that the different lines in Figure 4b were associated to all the plumes observed. Maybe the line width of each line could be coded as a function of the altitude? Obviously for dust plume one SD is significantly different from the others. Why ?

P15 L 388 : remove 'the' in front of plumes

P15 L 392-393 : This is impossible to see that on Figure 4. We could see something on Figure 4b but it could also be result of the CO concentration. . .

P15 L 395 – 396 : We must trust you about the fact that the smaller particles are observed within the lower dust plume. . .

P 15 L 401-406: Again you stated that below 2.5 km all the aerosol are mixed. You observed 3 dust plumes below 3km. I assume that these three samples where associated with the small particles. The BB plumes where observed above ∼2.2km. I'm not convinced that the location has anything to do but mainly the altitude. . . Most of the BB plumes have been observed close to big cities (Abidjan, Accra, Lome).

P15 L417-419 : The accumulation mode exceeds the amount of large particle. What is "large particle" here ? As there are no information about the mode the authors are referring to, it's hard to fallow. There is probably a need for a table that describe all the modes observed for each aerosol type. Anyway, is that surprising in number concentration?

P16 L 423. In this part you are talking about a phenomenon observed by colleagues during two specific days. Do they observe it during other days ? Could it happen during other days ? As you are not explaining the general mechanism there is no way for us to infer if this could have happen during the whole campaign. So when did you observed all plumes ? Was it during these specific days ? If not how do you know that this is always the case between the surface and 2.5km ?

P16 L430 : So the same dusty mode was observed during several campaigns over West Africa. From this observation you conclude that dust does not efficiently mix with other particles. Could you please explain ?

P16 L435 : Then, within the anthropogenic pollution plumes you observed a coarse mode. As the sites are nearby the coast you assume that it could be sea salt. Do you have any evidence for that? Could it be dust or road dust or pollens or any other large particles ???

P17 L458 : The strongest spectral dependence of SSA is observed for the lowest

absorption plume. Well from the Figure 5b this is not the case. Indeed, the largest SSA values for the BB aerosol at 440nm (Around 0.78) show a quite strong SSA at 660nm (around 0.74) and therefore quite a low spectral dependencies. The largest spectral depend are observed for both 'middle' SSA values at 440nm. Could you then remove the part right after that suggest similar results with the literature that are no more consistent with your results?

P18 L477 : I found it really odd for you to compare small SSA spectral dependencies observed over African cities using in-situ measurements with literature using AERONET measurements or even with plumes observed over European cities. The principle (column integrated versus in situ) of measurements as well as the pollutants (European polluted plumes are clearly different from African polluted plumes) have to be a bit similar to gain information. In this case arguing that what you observed is entirely different from these studies is not convincing.

P18 L 481 : During the NAMMA campaign, Omar et al. compare the extinction coefficients from the LIDAR and from the in situ measurements (assuming a dust LIDAR ratio). They found that the extinction was really close and then that no large aerosol was lost within the inlet. I'm sure you could do something similar here to estimate the losses for all cases.

P18 L484 : 'measurements are comparable to each other'. So I'd like to disagree with you. It depends on the event right ? Some events will be associated with uplifting of large particles and therefore the inlet cut off may play a major role OR some events could also be associated with uplifting of smaller particles and then the inlet cut-off won't be an issue anymore.

P19 clarify MEE versus MEC vs extinction mass efficiency vs extinction mass coefficient!

P19 : Another Mie test... So now you used a Mie code to estimate the impact of each aerosol mode ? Could you please explain what you did and how you did it ?

Âń We found MEC to be positively correlated with SAE (not shown) with measurement wavelengths (450–660 nm), which agrees with Mie theory. Âż Again, I don't understand the sentence. Are you talking about measured MEE vs measured SAE having the same Âń correlation Âż than the calculated one ? What about a straight comparison of SAE/MEE calculated and observed ? And how this Âń correlation Âż is relevant for your analysis ?

P19 L 509-511 : So the total and back-scattering coefficients have an error associated with the measurement principle. Did you correct the data fallowing Anderson and Ogren recommendations? What are the error associated with the measurements and then with the g calculation? Moreover, the g is also affected by the morphology of the particle.

P20 : Is this part 4 based on single SLR analysis ?

P20 L530 – 532 : On figure 6c there are no dots below 1500m and I cannot sea the NOx decrease that you are talking about.

P20 L 532 : You qualify it as a strong SSA variation (0.8 to 0.95) but since page 15 you told us that below 2.5km all the layers are mixed together. From the number concentration profile there are almost no dust at that altitude (below 1cm-3).

P21 L564 : Âń Figure 6 indicates markedly different processes affecting optical properties of biomass burning aerosols. Âż Which processes are highlighted by this figure ? To me, all BB profiles on Figure 6 are notably stable.

P21 L569 : Âń remote location Âż . Please define it. For me the anthropogenic plumes were also observed in remote location, don't you think ?

P21 L564 to P22 L 578 : I don't really understand the need for this paragraph. There are no new information and we also have to wait for further explanations.

P22 L589 – 591 Âń We did not find any correlation between the values of SSA and their spectral dependence, which suggests that the variability in SSA cannot be attributed

to different contributions of marine aerosol in pollution plumes. Âż So I'm not sure I do understand this statement. A correlation between SSA values and their spectral dependency is evidence that SS are not contributing much?

P22 L595 – 597 "The high absorbing properties (SSA∼0.81 at 550nm) of background aerosols is consistent with being a mixture of aged biomass burning and Atlantic marine aerosol. Âż Please explain.

And Âń Moreover SSA of background aerosol was lower than previously reported over the Southern Atlantic (Ascension Island) outside the fire season in Central Africa (Zuidem al., 2018), which supports this conclusion. Âż Now I'm missing something. You are comparing measurements performed over a small island, far from any anthropogenic sources or any natural sources of absorbing particles, to your results and found that aerosols you observed are more absorbing. Is that surprising at all ? Does that prove that you observed BB aerosols or rather anthropogenic aerosols ?

P22 L207 : A variation of SAE could be due to a size shift if you know that chemical composition, mixing state and morphology are similar. In this case you don't show any chemical results but you have the SD. Why using SAE to highlight the size variation ?

P23 L 611-613 Could you be more precise with the reference you cited ? Where were this studies perform and what were the particle diameter at the formation and after few hours ?

P23 L622 : Âń by the variability in composition of biomass burning aerosol Âż Please add a ref that prove that k is only linked with the aerosol chemical composition. Could you also show the spectral dependency of k ?

P25 L699 : You need to prove using chemical composition data that BB were not affected by the mixing with anthropogenic emissions (at least for the lowest SLR).

P26 L717 : You never show the background SSA spectral dependency.

P26 L 721 : Âń The large proportion of aerosols emitted from cities that resided in the

ultrafine mode particles" Are you talking about African Cities or any city in general ? Do you have a reference? From your observations, you always have the background aerosol mixed with the city plume, right ? So you cannot tell what is coming from cities and what is coming from the background, right ? Or you made a difference between both SD ?

The appendixes need to be called in the main text.

---

## Author Comment (AC1) · 6 Dec 2019

**Response to Referees**

**We thank the referees for taking the time to assess our manuscript and their useful suggestions for its improvement. Hereafter we answer all the comments.**

**Anonymous Referee #1**

General Comments

The authors present an overview of in-situ measurements of aerosol optical properties made during the DACCIWA aircraft measurement campaign over the coastal region of Southern West Africa in summer 2016. The region has a varied aerosol environment, in which they sampled mineral dust aerosol, biomass burning aerosol, and anthropogenic pollution. Along with the meteorological situation, instruments on the aircraft measured the aerosol size and vertical distributions, and the aerosol scattering, absorption and extinction coefficients. From this information the authors further derived the aerosol scattering and absorption Ångström exponents, the complex refractive index, the single scattering albedo (SSA), the mass extinction efficiency, and the asymmetry parameter. Using this information, the authors describe the vertical distribution of the aerosols with respect to their optical properties, as well as showing that the SSA of the measured biomass burning aerosols is primarily a function of the imaginary part of the refractive index rather than of the size distribution. The derived SSA values of the biomass burning aerosols appear to be particularly low compared to other measurements from other regions of the world.

This is a very straightforward paper, and I appreciate that the scope of this work is primarily concerned with presenting the measurements and providing an interpretation of their significance. Within this scope, I would say that this is a successful paper, and hence I only have a few minor comments to make.

Specific comments

Figure 1: I assume that the black lines are the flight tracks in background conditions?

**Black lines are the tracks of all the flights performed during the field campaign. They include flight tracks in background conditions as well as those in conditions that were not included in our classification due to various reasons (aerosol properties varying during a SLR, instruments not working properly,…).**

**We have modified Figure 1 to show the selected flight tracks in background conditions. We have also added the location of vertical profiles as requested by Referee #2.**

[Figure]

***Figure 1. Tracks of the 15 flights analyzed in this study. The colors indicate aircraft flight sampling layers dominated by biomass burning (green), mineral dust (orange), mixed dust-biomass***

*burning (red), anthropogenic pollution (blue) and background particles (grey) from both vertical profiles (squares) and straight and level runs (SLRs; thick lines).*

Section 2.2: as a qualitative comparison, I wonder if it would help to include one of these satellite images (MODIS or SEVIRI) over SWA with one of the flight tracks? This may be useful context to add and may aid the reader's understanding of the regional environment. Maybe such a figure may be more appropriate with respect to Section 2.3.2, and it may also be useful to have an image for each of the aerosol types.

**During the summer monsoon, large areas of southern West Africa and Atlantic Ocean are covered by clouds presenting a high surface albedo, making it challenging to observe aerosol properly from satellite observations. When analyzing MODIS or SEVIRI images, we were unable to follow the back-trajectories of dust or biomass burning plumes due to too many cloudy satellite pixels.**

Section 2.3.2: might it also be helpful to tabulate (briefly) the aerosol classification specifications? This may be helpful for quick reference when the reader wants to re mind themselves of how the aerosols are classified in the later sections and figures.

**The figure below has been added in the paper to illustrate how the aerosols were classified.**

[Figure]

*Figure 2. Absorption Ångström Exponent (AAE) as a function of the ratio NOx to CO. The markers are colored according to the Scattering Ångström Exponent (SAE). Classification of mineral dust, biomass burning and urban pollution particles has been added to the figure.*

Figure 5, labels: "wavelength"
**Done**

Anonymous Referee #2

This paper describes the aerosol physical and optical properties observed during the DACCIWA campaign. The paper is based on in-situ measurements performed a-board the ATR-42. The paper is well written and the topic fits the ACP's scope. However, I have many remarks and some needs to be taken into account before this paper could be published in ACP.

Major comments
The title of this manuscript. I believe the title refers to the BB aerosols that were observed with really low SSA in comparison to all previous results. However, this part is described only on the 23th page. I would strongly recommend changing it to "aerosol optical properties overview during the DACCIWA campaign" or something similar.
**The title has been changed to:** *"Overview of aerosol optical properties over southern West Africa from DACCIWA aircraft measurements"*

First major comments about the Mie code used for this study. So you stated few lines (P8 L237-245) about how you used the Mie code to retrieve the refractive index of aerosols (reverse method) based on their physical and optical properties. First of all you are using a Mie code for dust and BB aerosols. In fact you measure the asymmetry factor (which is not that good with the TSI nephelometer) and found out that the particles are far from spherical. You never stated the possible issues with it. Then you never say a word for the mixing state you used. I found a word about it later when you say that the mixing was external which was consistent with previous observations. Have you tested the case with internal mixing (either core shell or just coating) ? Moreover, what is the acceptable difference between measurements and calculation to stop the iterative process (5%, 10% more ??? )? Is it just iterative process with an a priori k and n or it an optimal estimation method that will avoid any local minimum? Later on (P20-21, L548-556), you used the Mie code again but this time I believe not in a reverse mode. So you entered a k and n along with the size distribution and you get optical properties that you used to calculate a SSA. Which k and did you used? In Africa, there are many sources of dust and BB aerosols associated with many different k and n... How these calculations made with a Mie code, for spherical particles, prove that your aerosols are externally mixed? You need here to prove it with a spheroidal code, for different mixing state.
**We have chosen to use a Mie code to calculate the aerosol optical parameters by assuming the particles are spherical. First, because the particle shape was not measured during the field campaign and it would be very hazardous to associate a shape to the aerosols, especially since aerosols from different sources were sampled at different times after emission. Second, this facilitates a quantitative comparison with past data, since a large majority of them used this simplification. Third, because climate models usually use Mie theory due its computational efficiency and applicability to radiative transfer models.**
**To explain this choice we added the following text in Section 2.3.1:** *"Optical calculations were performed using Mie theory, implying a sphericity assumption, because it facilitates a quantitative comparison with past data, mostly using this simplification and because most climate models assume spherical properties."*

**Concerning the aerosol mixing state, the application of Mie theory in our retrieval procedure to calculate aerosol optical parameters requires information about the size distribution, refractive index and particle density. These input parameters refer to the properties of the overall particle population within the aerosol. There is no assumption in the aerosol mixing state. By contrast, in climate models, the aerosol optical parameters are calculated from multiple particle types with**

different chemical composition and apply a mixing rule to calculate the refractive index of the aerosol from the refractive indices of the individual component species. Therefore, the mixing state of the aerosol must be addressed for climate modelling to calculate aerosol radiative forcing. Currently, the most common assumption made in climate models, which is also the easiest from a computational perspective, is that the aerosols are externally mixed. This is the reason why we assessed whether the assumption of external mixing allow to reproduce the observed optical properties in section 4.1. To test internal mixing assumption would need to know to what degree the aerosol population is internally-mixed. In fact, there are an enormous number of degrees of freedom in the aerosol mixing state. Even within a single distribution mode there are a prohibitively large number of combinations involved in mixing the constituents. At one extreme the aerosols can be internally mixed as a homogeneous material reflecting the chemical and physical average of all the contributing components. At another extreme, each aerosol component is physically separated from the other components creating an external mixture of chemically pure modes. The real mixing state can be expected to lie somewhere in between these two extremes, with aerosol in an external mode that is or is not an internal mixture of components. Therefore, in the absence of information on mixing state and chemical composition of individual particles, we cannot test internal mixing assumption.

We have added more information in the paper about the choice of the refractive indices used in the Mie code in section 4.1: *"The refractive indices at 550 nm were assumed to be 1.52-0.002i and 1.60-0.040i for dust and anthropogenic pollution particles, respectively, which are the mean values deduced from the data inversion procedure."*

We have also added more information about the iterative process developed to calculate aerosol optical parameters in section 2.3.1: *"The retrieval algorithm consists of iteratively varying the real part of the complex refractive index (m) from 1.33 to 1.60 and the imaginary part of the complex refractive index (k) from 0.000 to 0.080 in steps of resolution of 0.001. m and k were fixed when the difference between calculated values of $\sigma_{scat}$ and $\sigma_{abs}$ and measurements was below 1%."*

Number concentration of coarse particles. The authors are showing number concentrations of particles from 1-5um. These concentrations are measured from the OPC-GRIMM. The concentrations are between 0 and 2.5 cm-3. Could you provide the reader the noise of this instrument regarding these large particles. Is it significant ?

**The advantages of an OPC is that (1) it can count particle numbers very accurately when the concentration is low; (2) it has very good signal to noise ratios for large particles (>1μm). The figure below shows the time evolution of particle concentration from 1-5μm measured by the OPC-GRIMM for different aerosol plumes sampled during straight levelled runs. You can see that the signal to noise ratio was higher than 3 for all the aerosol plumes, which make the instrument well suited to quantify particle concentration in this size range.**

**We have added this information in section 2.3.1.: *"The signal to noise ratio of the OPC for particles in this size range was higher than 3, which makes the instrument well suited to quantify variations in $N_{coarse}$."***

[Figure]

*Temporal variation of particle number concentration measured in the size ranged 1-5µm by the GRIMM-OPC during different SLRs.*

During the field campaign, the ATR-42 performed vertical profiles at the beginning of each flight. These are used in Figure 3. On Figure 1, you highlighted the BB, dust and pollution plumes. How far are the vertical profiles from the plume observed ? I wonder how the variability of the vertical profile average is due to the distance to the actual plume. I had to wait until Figure 6 (P21) to have information about altitudes at which you observed the different plumes. At P14 L366-370, P15 L398-399, you are showing that aerosols, within a layer located below 2.5km, are a efficiently mixed with anthropogenic emissions. Then on Figure 6 you are trying to interpret the optical variations throughout the vertical profile. From my understanding, if you are interested in "pure" BB or Dust particles than you should only show/interpret vertical profiles from 2.5km up to 4km.

**The location of vertical profiles has been added in Figure 1. You can see that the vertical profiles were located near the straight leveled runs.**

[Figure]

*Figure 1. Tracks of the 15 flights analyzed in this study. The colors indicate aircraft flight sampling layers dominated by biomass burning (green), mineral dust (orange), mixed dust-biomass burning (red), anthropogenic pollution (blue) and background particles (grey) from both vertical profiles (squares) and straight and level runs (SLRs; thick lines).*

**Data presented in Figure 6 (now Figure 7 in the update manuscript) are based on measurements during SLR. This has been clarified in the paper at the beginning of section 4.1.: *"We exclusively***

*consider measurements acquired during SLRs, since only during these phases the whole set of aerosol optical properties were measured."*

**Minor remarks**

Page 2 L37 : I think something is missing to introduce properly the BB aerosols or just to skip a line here.

**The sentence has been rewritten as follows: "***For the layers dominated by the biomass burning particles, aerosol particles were significantly more light absorbing than those previously measured in other areas (e.g. Amazonia, North America) with SSA ranging from 0.71 to 0.77 at 550 nm.***"**

Page 2 L68 : It's usually called MEE that stands for Mass Extinction Efficiency and its units is m2/g not g/cm3. In the table2's caption you even called it extinction mass efficiency. Please be consistent throughout the manuscript. By the way how did you measure the mass concentration of aerosols ? Did you assume a density for each type ? And did you assume the same density for each mode ?

**We have changed MEC by MEE in the text.**

**Information about the mass densities of each aerosol type used to calculate aerosol mass concentration and henceforth MEC are already provided on P8 in L261-264.**

P3 L73-75: Why do you introduce remote sensing data that you will never ever talk again ?

**Our intention is to provide a comprehensive overview of aerosol optical properties over SWA for the climate modelling and the remote sensing communities. From the remote sensing perspective, in-situ data constitutes independent verification of satellite data products and can be used to assess the performance of inversion algorithms. For known TOA reflectance, the retrieval of AOD is very sensitive to aerosol size distribution and refractive index. Improper assumptions used in the algorithm regarding such key parameters may lead to a spurious AOD long-term trend in the regions under influence of absorbing particles due to the regional biases. This is the reason why we introduce remote sensing measurements in the introduction and we invite the remote sensing community to use this dataset in the conclusion.**

P3 L112 remove the 's' at aerosol

**Done**

P 5 When DACCIWA took place (month, year) ? What season was it (apparently during the monsoon period)?

**The campaign period is already presented before on P4.**

P6 L 176 : the CPC MARIE model needs to be described here. There is no paper associated with this instrument and I was unable to find any specifications on internet. Is it working with a flow regulator or a critical orifice? How is it counting in comparison to any TSI CPC ? You used the CPC measurements to infer the Nfine but you never say anything about a comparison, in term of number concentration, with the OPC, UHSAS or the SMPS. As this instrument is not (yet?) considered as a reference, you cannot use it like a reference. You could use the calibration data for the size distribution (PSL and oil) to prove that this instrument was measuring the same number concentrations than the OPC.

**The international standard method defined by the European Committee for Standardization for determining the particle number concentration is based on a CPC (CEN/TS 16976:2016). This instrument is considered as a reference for ambient aerosol monitoring for many years. The CPC Marie allows measuring number concentration of particle with a size above 10-15 nm**

(depending on the temperature conditions). It is structurally the same as a TSI CPC3010 except that there are more controls on the condensation and saturator temperatures and thermal dissipation has been improved. Thus, its measurement behavior is comparable to a TSI CPC3010 (Mertes, Schröder, and Wiedensohler, 1995; Russell et al., 1996; Wiedensohler et al., 1997). OPC, UHSAS and SMPS are measuring particle size distribution in limited size ranges (0.3-32µm for OPC, 0.1-1 µm for UHSAS during DACCIWA, 0.02-0.5 µm for SMPS). Given the relative importance of the smallest particles to the total concentration, the CPC concentration can be considered as the best estimation of the total particle concentration sampled.

We have added in the text the following references that provide a description and validation of the CPC Marie, as well as its intercomparison with other CPCs: *Mertes, Schröder, and Wiedensohler, 1995; Russell et al., 1996; Wiedensohler et al., 1997.*

Concerning the calibration experiment suggested by Referee #2, artificially produced PSL (or oil) particles would generate a mixture of air/water/PSL(oil) mixture with elevated concentration of ultrafine particles. These small particles cannot be fully detected by the OPC, UHSAS or SMPS instruments, so direct comparisons between instruments won't be relevant for this experiment. Otherwise, Marie CPC and SMPS CPC have been inter-compared in the laboratory at low pressure with concentration differences lower than 10%.

P6 L 187 Replace 525 by 550nm.
**Done**

P10 section 2.3.2: A figure showing the AAE versus the SAE would have been appreciated here.
**The new figure below has been added in the paper.**

[Figure]

*Figure 2. Absorption Ångström Exponent (AAE) as a function of the ratio NOx to CO. The markers are colored according to the Scattering Ångström Exponent (SAE). Classification of mineral dust, biomass burning and urban pollution particles has been added to the figure.*

P10 L282 : remove the 's' at the end of theses
**Done**

P11 L 301 : I believe that the background is also without the detectable influence of Dust and BB as well as anthropogenic emissions.
**Done**

P13 L334: The AEJ is a synoptic flow. It is always present over West Africa at a specific location (between 10-20 ◦N between 2-6km). The ATR flew between 3-11◦N. Therefore, I don't see how you would have been able to observe it and why you would have been able to observe it for specific cases and not for the others.

**The average climatological position of the AEJ varies during the summer between 10°N in June and 15°N in July. However, because it is a synoptic feature, the position of the AEJ core varies significantly from day to day, the DACCIWA period being no exception as shown by Knippertz et al., 2017 using ERA-I reanalysis data. In their Figure 8, Knippertz et al. (2017) show that the position of the AEJ on 2 July is centered at 4°N and on 10 July at 24°N. This is due to the many disturbances sweeping through southern West Africa during the monsoon season, which impact the location and the intensity of the AEJ core.**

**It is also worth pointing out the AEJ is a broad feature, generally confined to a width of 5°–10° of latitude as seen in reanalyses, but also in the airborne observations made using dropsondes along north-south transects across southern West Africa during JET 2000 (Tompkins et al., 2005) and AMMA in 2006 (Flamant et al., 2009). In the later study, the southern fringe of the AEJ is observed to reach the latitude of the SWA coast around mid-July.**

**Therefore, due to the important day to day variability of its mean position, but also of this latitudinal width, we can assess that observations made with the ATR between 3–11°N were indeed acquired in regions impacted by the AEJ and under a variety of AEJ-forcing conditions.**

**We have added a sentence about the latitudinal variation of the AEJ: "*Above 2.5 km amsl, the wind speed increases in the urban pollution and dust cases and the wind remains easterly, indicating the presence of the African easterly jet with its core typically farther north over the Sahel (Figure 8 in Knippertz et al., 2017 for the latitudinal variations of the African easterly jet during the DACCIWA field phase)."***

P13 L 335 : there are two dots after Sahel.
**Done**

P13 L356-357 : According to the Nfine vertical profile, the BB plume is observed from 2.5 – 4km. Your sentence isn't clear for me. Is this between the surface up to 4km or as suggested by the Nfine vertical profile ?
**We have clarified in the paper that we refer to the BB plumes transported above 1.5 km asl: "*The biomass burning plume extends from 1.5 to 5 km amsl and is associated with transport from the southwest in a layer of enhanced wind speed just below 4 km amsl as discussed above."***

Figure 4 : I think you could do something in this figure to show the altitude of each SLR. You never mentioned that the different lines in Figure 4b were associated to all the plumes observed. Maybe the line width of each line could be coded as a function of the altitude? Obviously for dust plume one SD is significantly different from the others. Why ?
**We have modified the figure to show the altitude of the aerosol sampled.**

[Figure]

**Figure 5. (a) Statistical analysis of number size distributions with colored areas representing the 3th, 25th, 75th and 97th percentiles of the data and (b) number size distributions normalized to CO for plumes dominated by biomass burning (green), dust (orange), mixed dust-biomass burning (red), anthropogenic pollution (blue) and background particles (black). In panel b, the line thickness is scaled by the altitude of the aerosol plume.**

P15 L 388 : remove 'the' in front of plumes
**Done**

P15 L 392-393 : This is impossible to see that on Figure 4. We could see something on Figure 4b but it could also be result of the CO concentration…
**Colors in the figure have been changed to enable a better observation of variability in particle concentration (see figure above).**

L 395 – 396 : We must trust you about the fact that the smaller particles are observed within the lower dust plume…
**We have added in the text that this observation can be seen in the previous Figure 3.**

P 15 L 401-406: Again you stated that below 2.5 km all the aerosol are mixed. You observed 3 dust plumes below 3km. I assume that these three samples where associated with the small particles. The BB plumes where observed above ∼2.2km. I'm not convinced that the location has anything to do but mainly the altitude… Most of the BB plumes have been observed close to big cities (Abidjan, Accra, Lome).
**As written in the paragraph, BB and dust plumes were sampled at approximatively the same altitude. The lowest BB plumes was observed at 2.2 km asl., which do not make a big difference with the dust plume sampled at 2.0 km asl. Since pollution plumes were exported up to 2.5 km asl., the altitude is probably not the reason of the different mixing with pollution plumes.**

P15 L417-419 : The accumulation mode exceeds the amount of large particle. What is "large particle" here ? As there are no information about the mode the authors are referring to, it's hard to fallow. There is probably a need for a table that describe all the modes observed for each aerosol type. Anyway, is that surprising in number concentration?
**"Larger particles" has been replaced by "particles centered at 230 nm" in the text for clarity.**

**It is already explained in the paragraph that the mode centered at 100 nm is surprising for aged biomass burning, when comparing with the literature.**

P16 L 423. In this part you are talking about a phenomenon observed by colleagues during two specific days. Do they observe it during other days ? Could it happen during other days ? As you are not explaining the general mechanism there is no way for us to infer if this could have happen during the whole campaign. So when did you observed all plumes ? Was it during these specific days ? If not how do you know that this is always the case between the surface and 2.5km ?

**We have removed the sentence referring to papers of Flamant et al. (2018) and Deroubaix et al. (2019).**

P16 L430 : So the same dusty mode was observed during several campaigns over West Africa. From this observation you conclude that dust does not efficiently mix with other particles. Could you please explain ?

**The coarse mode of the dust plume is expected to be impacted by the mixing with other particles in case of an internal mixing, which should somewhat increase the particle size. We have clarified it in the manuscript:** *"The super-micron mode of the dust plume is expected to be impacted by the mixing with other particles in case of an internal mixing, which should somewhat increase the particle size".*

P16 L435: Then, within the anthropogenic pollution plumes you observed a coarse mode. As the sites are nearby the coast you assume that it could be sea salt. Do you have any evidence for that? Could it be dust or road dust or pollens or any other large particles???

**Analysis of the spectral dependence of aerosol optical properties allows for interpretation of aerosol chemical composition. As stated in sections 2.3.1 and 2.3.2, mineral dust particles have the unique ability to absorb light strongly in the blue to ultraviolet spectrum. The presence of dust particles in aerosol plumes is thus expected to increase AAE above 1. We measured AAE in the range 0.7-1 in anthropogenic pollution plumes, which suggests negligible contribution of mineral dust in these plumes. This result is not surprising since anthropogenic pollution plumes originated from the south while the source region of dust was located in the north east when looking at the backtrajectories of air masses (Figure 2). For anthropogenic pollution plumes being sampled near the coast and below 0.5 km amsl, mixing of sea salt in pollution plumes is the most relevant interpretation for the presence of a coarse mode.**

**We have added the following explanation in the manuscript:** *"We measured AAE in the range 0.7-1 in anthropogenic pollution plumes (Figure 2), which suggests negligible contribution of mineral dust in these plumes. This coarse mode has most likely significant contributions from sea salt particles, as plumes arriving from the cities were transported at low altitude over the ocean (Fig. 3)."*

P17 L458 : The strongest spectral dependence of SSA is observed for the lowest absorption plume. Well from the Figure 5b this is not the case. Indeed, the largest SSA values for the BB aerosol at 440nm (Around 0.78) show a quite strong SSA at 660nm (around 0.74) and therefore quite a low spectral dependencies. The largest spectral depend are observed for both 'middle' SSA values at 440nm. Could you then remove the part right after that suggest similar results with the literature that are no more consistent with your results?

**The referee is right. We have removed this part.**

P18 L477 : I found it really odd for you to compare small SSA spectral dependencies observed over African cities using in-situ measurements with literature using AERONET measurements or even with plumes observed over European cities. The principle (column integrated versus in situ) of measurements as well as the pollutants (European polluted plumes are clearly different from African polluted plumes) have to be a bit similar to gain information. In this case arguing that what you observed is entirely different from these studies is not convincing.

**AERONET has been extensively compared with various types of airborne in situ airborne measurements for a range of different aerosol types including urban pollution particles and the Africa continent. These were recently summarized in Andrews et al. (2017). They confirm previously reported comparison that the large majority of SSA comparisons for AOD> 0.2 are within the reported in-situ SSA uncertainty bounds, while AERONET inversions tend to overestimate SSA for low AOD. All AERONET measurements that included the AOD > 0.2 constraint have shown that SSA systematically decrease with increasing wavelength for a range of different urban pollution plumes across the world (Dubovick et al., 2002; Shin et al., 2019). However, in the light that no AERONET or in-situ measurements are available for comparison in other African cities, we have nuanced the sentence:** *"However, the flat spectral dependence of SSA appears to be anomalous for anthropogenic pollution aerosols, as SSA has been shown to decrease with increasing wavelength for a range of different urban pollution plumes over European, American and Asian cities (Dubovick et al., 2002; Di Biagio et al., 2016; Shin et al., 2019)."*

P18 L 481 : During the NAMMA campaign, Omar et al. compare the extinction coefficients from the LIDAR and from the in situ measurements (assuming a dust LIDAR ratio). They found that the extinction was really close and then that no large aerosol was lost within the inlet. I'm sure you could do something similar here to estimate the losses for all cases.

**This paragraph was unclear. We wanted to highlight the difficulty to compare our results with previous measurements because the measurement of SSA is highly dependent on the extent to which the coarse mode is measured behind the aerosol sampling inlet. For instance, Ryder et al. (2013) found an overestimation of SSA in dust plumes when the SSA is measured behind inlet during the FENNEC campaign. In our case, the impact of the Community Aerosol Inlet on the measured optical properties has been calculated for dust aerosols in Denjean et al. (2016). We found that the absolute error associated with SSA, g and MEE due to the aircraft inlet is in the range covered by the measurement uncertainties. However, different aerosol inlet systems were used during previous field campaigns (cut-off diameter of 5µm during DACCIWA and 2.5µm for Weinzierl et al. (2011) during SAMUM-2 for instance), which can lead to discrepancies when comparing SSA from different campaigns.**

**The paragraph has been changed as follows:** *"The magnitude of SSA increased at the three wavelengths when dust events occurred. Large variations in SSA were obtained with values ranging from 0.76–0.92 at 440 nm, 0.81–0.94 at 550 nm and 0.81–0.97 at 660 nm. The measurement of SSA is highly dependent on the extent to which the coarse mode is measured behind the aerosol sampling inlet. Denjean et al. (2016) found that the absolute error associated with SSA, g and MEE of dust aerosols due to the CAI inlet is in the range covered by the measurement uncertainties. However, different aerosol inlet systems were used during previous field campaign, which makes comparison of our results with previous measurements difficult."*

P18 L484 : 'measurements are comparable to each other'. So I'd like to disagree with you. It depends on the event right ? Some events will be associated with uplifting of large particles and therefore the inlet cut off may play a major role OR some events could also be associated with uplifting of smaller particles and then the inlet cut-off won't be an issue anymore.

**As explained in the previous comment, the paragraph was unclear and has been rewritten.**

P19 clarify MEE versus MEC vs extinction mass efficiency vs extinction mass coefficient!

**We have exchanged MEC by MEE in the text.**

P19: Another Mie test… So now you used a Mie code to estimate the impact of each aerosol mode ? Could you please explain what you did and how you did it ? Ân We found MEC to be positively correlated with SAE (not shown) with measurement wavelengths (450–660 nm), which agrees with Mie theory. Âz Again, I don't understand the sentence. Are you talking about measured MEE vs measured SAE having the same Ân correlation Âz than the calculated one ? What about a straight comparison of SAE/MEE calculated and observed ? And how this Ân correlation Âz is relevant for your analysis ?

**This is not another Mie test; this is the result of the iterative process that provides SSA, MEE and g. The aim of the sentence is to explain some of the variability in MEE for a given aerosol type. SAE being a good proxy for variation of size distribution for a given aerosol chemical composition, it was expected that MEE correlates with SAE.**

**To clarify this, we have rewritten the sentences as follows:** *"MEE is heavily influenced by the mass concentrations in the accumulation mode where the aerosol is optically more efficient in extinguishing radiation. We found MEE to be positively correlated with SAE (not shown), which was expected because of the dependence of MEE on particle size".*

P19 L 509-511: So the total and back-scattering coefficients have an error associated with the measurement principle. Did you correct the data fallowing Anderson and Ogren recommendations? What are the error associated with the measurements and then with the g calculation? Moreover, the g is also affected by the morphology of the particle.

**Data were corrected for truncation error using a Mie code in the data inversion procedure that takes into account the aerosol size distribution. This correction is more accurate than the Anderson and Ogren recommendations which are based on single sample. This was already written in Table 1 and Appendix 2, but we have added the following sentence in section 2.1. :** *"The particle scattering coefficients ($\sigma_{scat}$) at 450, 550 and 635 nm were measured using a AURORA3000 3-wavelength nephelometer and corrected for angular truncator error in the data inversion procedure using a Mie code (i.e. Section 2.3.1 and Appendix 2)."*

**Concerning the aerosol morphology, please look at our response above in major comments.**

P20: Is this part 4 based on single SLR analysis ?

**Yes, it is now specified in the text:** *"We exclusively consider measurements acquired during SLRs, since only during these phases the whole set of aerosol optical properties were measured."*

P20 L530 – 532 : On figure 6c there are no dots below 1500m and I cannot sea the NOx decrease that you are talking about.

**The sentence has been removed.**

P20 L 532 : You qualify it as a strong SSA variation (0.8 to 0.95) but since page 15 you told us that below 2.5km all the layers are mixed together. From the number concentration profile there are almost no dust at that altitude (below 1cm-3).

**We clearly mentioned previously that large variations in the concentration of fine particles (sections 3.1 and 3.2) and SSA (section 3.3) were observed in dust plumes. Concerning the concentration, only small number concentration of dust particles will contribute significantly to the aerosol mass concentration and extinction coefficient, and consequently to the aerosol radiative effect. You can see that effect in Fig 3a, which shows that the extinction coefficient increased from 20 to 80 Mm$^{-1}$ when dust was transported at 2.5 km asl over the region.**

P21 L564 : Án Figure 6 indicates markedly different processes affecting optical properties of biomass burning aerosols. Âz Which processes are highlighted by this figure ? To me, all BB profiles on Figure 6 are notably stable.
**The sentence has been removed.**

P21 L569 : Ân remote location Âz . Please define it. For me the anthropogenic plumes were also observed in remote location, don't you think ?
**We have removed the part of the sentence about the location of BB.**

P21 L564 to P22 L 578 : I don't really understand the need for this paragraph. There are no new information and we also have to wait for further explanations.
**The important message of this paragraph is that BB plumes were not affected by pollution emissions. This interpretation is based on Fig. 6 showing the vertical distribution of SSA, SAE and NOx concentrations. So we think it is very important to keep this discussion here. However, we agree that the end of the paragraph can be moved in section 4.2 and it has been done.**

P22 L589 – 591 Ân We did not find any correlation between the values of SSA and their spectral dependence, which suggests that the variability in SSA cannot be attributed to different contributions of marine aerosol in pollution plumes. Âz So I'm not sure I do understand this statement. A correlation between SSA values and their spectral dependency is evidence that SS are not contributing much?
**We have removed this sentence.**

P22 L595 – 597 "The high absorbing properties (SSA□0.81 at 550nm) of background aerosols is consistent with being a mixture of aged biomass burning and Atlantic marine aerosol. Âz Please explain. And Ân Moreover SSA of background aerosol was lower than previously reported over the Southern Atlantic (Ascension Island) outside the fire season in Central Africa (Zuidem al., 2018), which supports this conclusion. Âz Now I'm missing something. You are comparing measurements performed over a small island, far from any anthropogenic sources or any natural sources of absorbing particles, to your results and found that aerosols you observed are more absorbing. Is that surprising at all ? Does that prove that you observed BB aerosols or rather anthropogenic aerosols ?
**Concerning the interpretation of the low SSA in background plumes, we have completed the sentence as follows:** *"The high absorbing properties (SSA~0.81 at 550nm) and the presence of particles both in the accumulation and super-micron modes (i.e. section 3.2.) in background plumes are consistent with being a mixture of aged absorbing biomass burning and Atlantic marine aerosol."*
**We have removed the sentence comparing our measurements with those of Zuidema et al. (2018), since it is discussed in more details in the next section.**

P22 L207 : A variation of SAE could be due to a size shift if you know that chemical composition, mixing state and morphology are similar. In this case you don't show any chemical results but you have the SD. Why using SAE to highlight the size variation ?

**As mentioned in section 2.3.1., the value of SAE depends primarily on the size of the particles, such that small values of SAE indicate larger aerosol particle, and large values of SAE indicate relatively smaller aerosol. This parameter is widely used in the community for studying the variability in aerosol size distribution because it is a sensitive parameter that needs limited corrections (only the truncation of the nephelometer). Using the measured aerosol size distribution would bring a higher degree of uncertainty due to the need to correct measurements with the aerosol refractive index.**

L 611-613 Could you be more precise with the reference you cited ? Where were this studies perform and what were the particle diameter at the formation and after few hours ?
**We have completed the sentence as follows: "*with previous observations of size distribution in aged North American biomass burning plumes*". The range of particle diameter reported in these studies was already provided section 3.2.**

P23 L622 : Ân by the variability in composition of biomass burning aerosol Âz Please add a ref that prove that k is only linked with the aerosol chemical composition. Could you also show the spectral dependency of k ?
**k is not only linked with the aerosol chemical composition, it also depends on the size distribution of the particle species. However, given that SSA was previously found to be independent of the aerosol size distribution (Figure 8a), Figure 8b suggests that SSA variability was strongly influenced by the variability in composition of biomass burning aerosol. The paragraph has been rewritten as follows: "*In contrast, Figure 8b shows that there was a consistent decrease in SSA with increasing k, although there is some variability between the results from different plumes. The observed variability of SSA is reflected in a large variability of k, which is estimated to span the large range 0.048–0.080 at 550 nm. k depends both on the aerosol chemical composition and size distribution (Mita and Isono, 1980). Given that SSA was found to be independent of the aerosol size distribution (Figure 8a), Figure 8b suggests that SSA variability was strongly influenced by the variability in composition of biomass burning aerosol, implying a high contribution for light-absorbing particles.*"**

P25 L699 : You need to prove using chemical composition data that BB were not affected by the mixing with anthropogenic emissions (at least for the lowest SLR).
**This has been intensively discussed in section 4.1 by looking at the vertical variability of SSA, SAE and NOx concentration. It was also discussed in section 3.2 with the analysis of aerosol size distribution in BB plumes.**

P26 L717 : You never show the background SSA spectral dependency.
**It is already shown Fig. 5 (a) in grey.**

P26 L 721 : Â The large proportion of aerosols emitted from cities that resided in the ultrafine mode particles" Are you talking about African Cities or any city in general ? Do you have a reference? From your observations, you always have the background aerosol mixed with the city plume, right ? So you cannot tell what is coming from cities and what is coming from the background, right ? Or you made a difference between both SD ?
**The interpretation of the ultrafine mode has been already intensively discussed in section 3.2. As stated in the text, this mode is only observed in plumes originating from the polluted cities of Lomé, Accra and Abidjan.**

In the conclusion, we have clarified the origin of the anthropogenic aerosols: *"The large proportion of aerosols emitted from the cities of Lomé, Accra and Abidjan that resided in the ultrafine mode particles have limited impact on already elevated amounts of accumulation mode particles having a maximal absorption efficiency"*.

The appendixes need to be called in the main text.
**Appendixes were already called in the text (P5 for Appendix 1 and P7 for Appendix 2).**

[revised manuscript text omitted]

---

## Editor Decision (ED1)

acp-2019-587 Decision

Dear Cyrielle,

my apologies for the delay in the decision on the manuscript. I expected feedback from one reviewer who raised major concerns but never received an answer. So I conducted an editor's review of the revised manuscript which, however, took some time – sorry.

I request minor changes of mostly technical nature, which are listed in the following. Once I have received the revised manuscript, the decision will be made immediately.

Line numbers refer to the annotated manuscript.

**GENERAL**

Please check for NO$_x$ instead of NOx.

Please check for the correct use of singular and plural cases.

I am wondering why the overview paper on dust properties by Formenti et al. (2011) is not referenced in the manuscript.

**DETAILED**

Line 23: please correct: "… as part of the …"

Line 71: MEE should be introduced as "mass extinction efficiency"

Line 165 – 167: Referring to a comment of Reviewer 2, I suggest adding bit more explanatory information on the CPC MARIE since this instrument is not documented in literature. I propose something like "butanol-based conductive cooling type CPC" to give technical information.

Line 170: Please add information about the size range covered by the GRIMM OPC.

Line 178: Please rephrase: "TSI 3563 3-Wavelength Integrating Nephelometer" and refer to "integrating nephelometer" in the text.

Line 190 – 191: It would be beneficial to quantify the agreement between CAPS PM$_{ex}$ and NEPH + PSAP. Is the agreement within the range reported by Petzold et al. (2013) for the same instrument configuration?

Line 233: I am a bit confused with the nomenclature used for the complex refractive index. To my knowledge, the agreed terminology is " m = n + $i$ k" but I understand that you refer to the real part as "m".

Line 255: can you please add a range of values used for MEE in your study? In the current version, the reader gets the impression that MEE is given in g/cm$^3$.

Line 283: AAE for biomass burning aerosol can reach values close to 2.0, see Sandradewi et al. 2008. Please include this into your discussion.

Figure 4: I suggest modifying the y-axis title to "Altitude [m amsl]. Then, it is consistent with the nomenclature used in the text.

Line 399: To stay consistent with the rest of the manuscript, I suggest rephrasing "… , large differences were observed, … ".

Line 490 ff:  compilation of SSA values observed in the complex mixture of mineral dust and anthropogenic emissions over the city of Dakar is reported in Petzold et al. (2011). Here, we also found strong wavelength dependence of SSA for anthropogenic emissions, using the same instrumentation you have used in your study. It might be worthwhile referring to these observations here.

Table 2: Please refer here also to mass extinction efficiency (MEE) instead of extinction mass efficiency. Please mention units for the properties listed in Table 2.

Line 542: Please correct: "…the whole sets of …"

Line 562: Please correct "… dust size distributions …"

Figure 7: Please adjust font size of x-axis labels; currently this is hard to read.

Line 614: Please correct: "The high CO values … further indicates …"

Appendix 2: I suggest using CAPS $PM_{ex}$ for the instrument used here, to distinguish from CAPS $NO_2$ and CAPS $PM_{SSA}$. Please adjust throughout the whole manuscript.

**REFERENCES**

Formenti, P., Schütz, L., Balkanski, Y., Desboeufs, K., Ebert, M., Kandler, K., Petzold, A., Scheuvens, D., Weinbruch, S., and Zhang, D.: Recent progress in understanding physical and chemical properties of African and Asian mineral dust, Atmos. Chem. Phys., 11, 8231-8256, doi: 10.5194/acp-11-8231-2011, 2011.

Petzold, A., Veira, A., Mund, S., Esselborn, M., Kiemle, C., Weinzierl, B., Hamburger, T., Ehret, G., Lieke, K., and Kandler, K.: Mixing of mineral dust with urban pollution aerosol over Dakar (Senegal): impact on dust physico-chemical and radiative properties, Tellus, 63B, 619-634, doi: 10.1111/j.1600-0889.2011.00547.x, 2011.

Petzold, A., Onasch, T., Kebabian, P., and Freedman, A.: Intercomparison of a Cavity Attenuated Phase Shift-based extinction monitor (CAPS PMex) with an integrating nephelometer and a filter-based absorption monitor, Atmos. Meas. Tech., 6, 1141-1151, doi: 10.5194/amt-6-1141-2013, 2013.

Sandradewi, J., Prevot, A. S. H., Weingartner, E., Schmidhauser, R., Gysel, M., and Baltensperger, U.: A study of wood burning and traffic aerosols in an Alpine valley using a multi-wavelength Aethalometer, Atmos. Environ., 42, 101-112, doi: 10.1016/j.atmosenv.2007.09.034, 2008.

---

## Author Response (AR2)

Dear Dr. Andreas Petzold,

We want to thank you for the review of the manuscript. Based on your comments, we have made revisions to the original manuscript as follows:

- Papers of Sandradewi et al. (2008), Formenti et al (2011), Petzold et al. (2013) have been incorporated.
- Technical information on the CPC, UHSAS, GRIMM and SMPS have been added.
- A brief comparison of AAE in biomass burning plumes with literature has been included.
- Figure 4, Figure 7 and Appendix 2 have been improved.
- A number of typographical corrections have been made to the text.

We hope that these changes will fulfill your suggestions. The article with tracked changes can be found below.

Yours sincerely, Cyrielle Denjean

**1 Overview of aerosol optical properties over southern**

**2 West Africa from DACCIWA aircraft measurements**

Cyrielle Denjean1, Thierry Bourrianne1, Frederic Burnet1, Marc Mallet1, Nicolas

- 5 Maury1, Aurélie Colomb2, Pamela Dominutti2,\*, Joel Brito2,\*\*, Régis Dupuy2,
- 6 Karine Sellegri2, Alfons Schwarzenboeck2, Cyrille Flamant3, Peter Knippertz4
- 7

1CNRM, Université de Toulouse, Météo-France, CNRS, Toulouse, France

2LaMP, Université de Clermont Auvergne, Clermont-Ferrand, France

- 3LATMOS/IPSL, Sorbonne Université, UVSQ, CNRS, Paris, France

[revised manuscript text omitted]